# Probing Neural Combinatorial Optimization Models

**Zhiqin Zhang**
Singapore Management University
`zqzhang.2020@phdcs.smu.edu.sg`

**Yining Ma**
Massachusetts Institute of Technology
`yiningma@mit.edu`

**Zhiguang Cao**
Singapore Management University
`zgcao@smu.edu.sg`

**Hoong Chuin Lau**[*]
Singapore Management University
`hclau@smu.edu.sg`

## Abstract

Neural combinatorial optimization (NCO) has achieved remarkable performance, yet its learned model representations and decision rationale remain a black box. This impedes both academic research and practical deployment, since researchers and stakeholders require deeper insights into NCO models. In this paper, we take the first critical step towards interpreting NCO models by investigating their representations through various probing tasks. Moreover, we introduce a novel probing tool named Coefficient Significance Probing (CS-Probing) to enable deeper analysis of NCO representations by examining the coefficients and statistical significance during probing. Extensive experiments and analysis reveal that NCO models encode low-level information essential for solution construction, while capturing high-level knowledge to facilitate better decisions. Using CS-Probing, we find that prevalent NCO models impose varying inductive biases on their learned representations, uncover direct evidence related to model generalization, and identify key embedding dimensions associated with specific knowledge. These insights can be potentially translated into practice, for example, with minor code modifications, we improve the generalization of the analyzed model. Our work represents a first systematic attempt to interpret black-box NCO models, showcasing probing as a promising tool for analyzing their internal mechanisms and revealing insights for the NCO community. The source code is publicly available [2].

## 1 Introduction

Neural combinatorial optimization (NCO) has demonstrated remarkable performance in solving classic combinatorial optimization problems, such as vehicle routing, achieving results comparable to, or even surpassing, specialized heuristic algorithms such as Concorde [1], ACO [2], LKH3 [3], HGS [4]. However, the underlying reasons behind these impressive results, particularly the nature of the knowledge learned by these neural models, remain largely unexplored and unclear.

In this paper, we take the first step towards interpreting NCO models by *probing* [5, 6, 7], a powerful tool that has proven successful in computer vision (CV) and natural language processing (NLP). We pioneer the first study that directly investigates the learned embeddings in deep NCO models, aiming to explore the internal mechanisms of these black-box models. In this work, we aim to address two fundamental questions regarding the representations learned by NCO models: (i) What decision-related knowledge do they acquire? (ii) How do they learn and utilize this knowledge?

---

[*]Corresponding author

[2]Source Code: `https://github.com/123zhangzq/NeurIPS2025_probing/`

39th Conference on Neural Information Processing Systems (NeurIPS 2025).

Addressing the First Question: Probing involves training auxiliary prediction tasks using the embeddings learned by a trained deep learning model. In the context of NLP, for instance, if a simple model (particularly a linear model) can be trained to predict linguistic information about a word (e.g., its part-of-speech tag) or a pair of words (e.g., their semantic relation) from the embeddings, we can reasonably conclude that the embeddings successfully encode this information (see [8] for more details). However, unlike NLP tasks, which naturally involve intuitive subtasks that are well-suited for probing, combinatorial optimization (CO) problems typically lack such directly applicable subtasks. To address this gap, we systematically design a set of probing tasks specifically aimed at exploring both low- and high-level decision-related knowledge within NCO models.

Addressing the Second Question: Understanding how deep learning (DL) models learn and utilize decision-related knowledge remains a challenging problem. Different approaches have been proposed, such as using probing to explore decision boundaries [9] to understand in-context learning in large language models (LLMs), or identifying individual neurons whose input or output weights have high cosine similarity with the learned probe direction to detect specific neurons that encode particular knowledge [10]. In this paper, we propose a novel method called Coefficient Significance Probing (CS-Probing) to investigate the representations of deep models. CS-Probing not only identifies the most informative embedding dimensions but also quantitatively assesses their statistical significance, providing deeper insights into the model's decision-making process.

**Contributions.** Our main contributions are as follows: **(1)** We systematically design probing tasks for NCO models and demonstrate that their representations capture both low-level decision-related information (e.g., perceiving Euclidean distances) and high-level knowledge (e.g., avoiding myopic decisions based solely on distance) for decision-making. **(2)** Through our proposed CS-Probing, we discover that different NCO models introduce diverse inductive biases into the learned representations, resulting in varied decision-making patterns. **(3)** By applying CS-Probing, we identify the key embedding dimensions that encode specific knowledge within the model representations. **(4)** Leveraging these key dimensions, we provide evidence of the generalization capabilities of NCO models: models with superior generalization consistently utilize the same embedding dimensions across different tasks. In contrast, models whose knowledge becomes disorganized across dimensions during generalization tend to experience performance degradation. Finally, based on probing insights, we show that modifying only a few lines of code in an NCO model holds the potential to improve its generalization performance, illustrating the practical value of our proposed probing analysis.

## 2    Preliminaries

**NCO model.** NCO models are a class of learning-based solvers for CO problems, with more details provided in Appendix A.1. This paper studies the most representative transformer-based architectures (as illustrated in Figure 1) as examples to demonstrate how probing can explore their internal mechanisms. Taking the Traveling Salesman Problem (TSP) as an example, the raw features of nodes (e.g., xy coordinates in the Euclidean space) are first fed as inputs. These raw features are projected into a high-dimensional space through a linear projection layer. Subsequently, multiple attention mechanism layers are employed to integrate abstract features from different nodes, yielding the final representations for each node. These representations are then processed either through a compatibility calculation or directly projected to a scalar via a linear transformation, followed by a softmax operation to obtain the selection probability of the next node. Recursively, this process connects all nodes to generate the complete TSP solution. In this paper, we utilize probing to investigate the representations of three models: AM [11], POMO [12], and LEHD [13].

**Probing.** We use a linear probing [5, 14, 8, 15, 16, 17] to explore the representations of NCO models. If this linear model can accurately predict the probing tasks based on the embeddings from the NCO models, it indicates that the knowledge relevant to the probing tasks can be easily extracted from the embeddings [5, 8]. This also suggests that the pre-trained NCO model, from which the embeddings for the probing tasks are derived, has the ability to encode this knowledge in its representations. Other interpretability approaches, including gradient-based attribution, visualization, and neuron ablation, lack the explanatory power of probing. For example, the first method reveals output sensitivity to input features but neither shows where or whether knowledge is encoded in hidden representations nor provides structural insight into how the model organizes the solution space. In contrast, probing offers a systematic interpretability framework by assessing the linear decodability of target properties

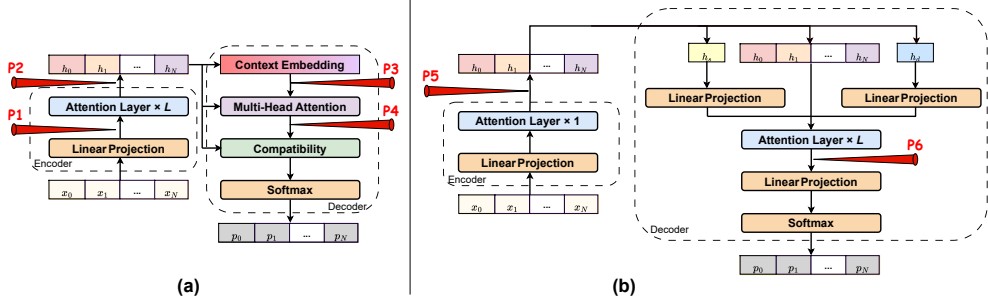

Figure 1: Two groups of NCO model architectures: (a) represents the HELD structure, as seen in AM and POMO, while (b) represents the LEHD structure. The red arrows in the figure indicate the positions where we probe the model, extracting the embeddings.

from intermediate representations. For a comprehensive discussion of the diverse applications of probing, the reader is referred to Appendix A.2.

# 3 Probing for NCO

## 3.1 Probing Tasks and Setup

Since combinatorial optimization problems do not have suitable subtasks to serve as probing tasks, new task design is necessary for the specific CO problem being explored. Using the TSP problem as an example, we propose two probing tasks to investigate NCO models: whether the model can perceive the Euclidean distance between nodes (*Probing Task 1*); and whether the model can learn to avoid constructing solutions in a myopic manner, such as greedily connecting to the nearest node (*Probing Task 2*). The former can be regarded as low-level features of the TSP problem, while the latter is relatively higher-level and more closely aligned with the final decision-making process.

We also introduce two probing tasks for the CVRP problem, examining whether NCO models can understand constraints (*Probing Task 3* and *Probing Task 4*). *Probing Task 3* investigates a relatively simple low-level feature: the linear additive relationship between node demands. In contrast, *Probing Task 4* examines a higher-level property: whether the embeddings encode information about two nodes belonging to the same route in the optimal solution.

Figure 7 in Appendix illustrate examples of probing dataset creation and label collection. For detailed definitions of these four probing tasks, the integration of domain knowledge from the CO field to create datasets, and the embeddings as probing input, please refer to Appendix B.

## 3.2 Probing the Decision-Related Knowledge in NCO Models

### Can NCO learn decision-related information from routing problems, exemplified by TSP?

We first examine whether NCO models capture a key low-level feature of TSP: the Euclidean distance between nodes. Next, we explore a higher-level and more abstract aspect of the embeddings: whether NCO models can capture decision-related information that avoids the myopic strategy of always selecting the nearest node. Having established the presence of both low- and high-level features, we perform a layer-wise analysis to understand how such information is encoded and learned during training. This section summarizes key findings in Table 1. In this table, "w/o ints" and "w/ ints" denote the absence and presence

Table 1: Highlights from Table 9. Here, we conduct the evaluation process 10 times to report the mean ± SEM.

|  | Probing input | Task 1 ($R^2$) | Task 2 (AUC) |
|---|---|---|---|
| w/o ints. | AM-Init | -0.0003 ± 0.00000 | 0.49 ± 0.00 |
|  | AM-Enc-$l3$ | 0.2529 ± 0.00048 | 0.76 ± 0.00 |
|  | POMO-Enc-$l6$ | 0.1981 ± 0.00001 | 0.76 ± 0.00 |
|  | LEHD-Dec-$l6$ | 0.9418 ± 0.00031 | 0.86 ± 0.00 |
| w/ ints. | AM-Init | 0.7111 ± 0.00000 | 0.52 ± 0.00 |
|  | AM-Enc-$l3$ | 0.9282 ± 0.00035 | 0.83 ± 0.00 |
|  | POMO-Enc-$l6$ | 0.7917 ± 0.00000 | 0.86 ± 0.00 |
|  | LEHD-Dec-$l6$ | 0.9415 ± 0.00027 | 0.86 ± 0.00 |

of interaction terms (see Appendix B.2.2), while the "Probing input" columns, 'Enc-lx' and 'Dec-lx' refer to the x-th layers in the encoder and decoder of NCO models. Complete results and discussion are presented in Appendix C.1.1, with Table 9 providing a comprehensive summary.

***Probing Task 1*: Euclidean distance.** We examine the ability of the three NCO models to linearly represent the Euclidean distances between pairs of nodes (specifically, the current node and any unvisited node) during decision-making, by training linear probes and evaluating their performance.

As shown in the "w/o ints." rows of AM-Init in Table 1 for both 20-node and 100-node examples, the values indicate that the initial embeddings of AM fail to capture the nonlinear relationship of Euclidean distance (with $R^2$ values close to 0). These embeddings are derived by mapping the raw features, specifically the 2D coordinates in the TSP, through a linear projection into the shared dimensional space of the encoder and decoder (128 dimensions for all three models discussed in this paper). In "Knowledge existence" section of Appendix C.1.1, we explain that the $R^2$ of a Euclidean distance regression model using node coordinates as input is zero because it cannot capture the nonlinear nature of Euclidean distance. Thus, the initial embeddings essentially retain the properties of the raw features and similarly fail to linearly capture Euclidean distances. The phenomenon of an $R^2$ value of zero for the initial embeddings can be observed across all NCO models.

However, after passing through the NCO model, the $R^2$ values for AM, POMO, and LEHD increase to 0.2529, 0.1981, and 0.9421, respectively, for 20-nodes example, as shown in Table 1. Moreover, when considering interaction terms, the $R^2$ values for all three models' embeddings after the encoder or decoder are significantly higher than those of the initial embeddings, approaching 1. This indicates that the representations in these NCO models contain linearly decodable Euclidean distance information, meaning they have learned how to linearly represent Euclidean distances.

***Probing Task 2*: Avoidance of myopia.** Through *Probing Task 2*, we explore whether NCO models can learn to avoid making decisions based solely on distance. As a first step, we train a probing model using raw path feature (distance) as input. The resulting performance, with an AUC close to 0.5, confirms that *Probing Task 2* is not merely a trivial path discrimination task.

We use the "AM-Init" results as a baseline reference, with AUC consistently at 0.5, indicating that the initial embeddings cannot linearly extract the knowledge needed to distinguish which nodes are connected to the current ones in the global optimal solutions (namely, the optimal edges). To confirm that *Probing Task 2* is not relying on Euclidean distances for node differentiation, we further examine the initial embeddings with interaction terms, whose AUC values remain close to 0.5, suggesting they still fail to distinguish between optimal or greedy edges. In contrast, in *Probing Task 1*, the initial embeddings with interaction terms achieve an $R^2$ above 0.7, indicating that the initial embeddings with interaction terms have linear explanatory power for Euclidean distances. This observation confirms that the two probing tasks are fundamentally different. It also implies that if the embeddings in an NCO model can be linearly distinguished in *Probing Task 2*, the model has learned to avoid myopic decision-making and capture the knowledge needed to find the global optimal solution.

The results in Table 1 demonstrate that all three NCO models exhibit the ability to avoid myopic decision-making, with AUC scores exceeding 0.8. Notably, on both 20-node and 100-node instances, this ability is aligned with three models' performance on the optimization problem outcomes, as discussed in Section 3.3, where we analyze the relationship between probing performance and final model performance. Moreover, this ability is consistently stronger in the final layer compared to the first layer for all three models. In the next section, we provide a more fine-grained analysis to illustrate how the behavior of NCO models evolves across layers and how it changes during training.

**Fine-grained analysis.** In Figure 2, we present the results of two TSP probing tasks across different layers of embeddings for three trained models. The observation shows that the initial embeddings (before any attention layers) exhibit weak Euclidean distance perception. However, after just one attention layer, all models achieve strong distance awareness, which slightly weakens with depth. Despite this, deeper layers help NCO models develop high-level decision-making abilities, such as avoiding myopic node choices. An exception occurs in the final layers of LEHD, where this ability slightly declines, possibly due to the emergence of more complex strategies. For an analysis of NCO model layers and their varying trends in capturing low- and high-level knowledge with increasing layers, see the "Results by Model Layer" section in Appendix C.1.1. Overall, NCO models transition from learning spatial relations in shallow layers to strategic reasoning in deeper layers.

Figure 3 illustrates the evolution of results for two TSP probing tasks during the training process. As shown, for AM and POMO, the model performance improvement during the initial epochs is the fastest, and the results for *Probing Task 2* (related to avoiding myopic decisions) also improve rapidly in early learning epochs. In contrast, LEHD achieves peak performance on *Probing Task 2*

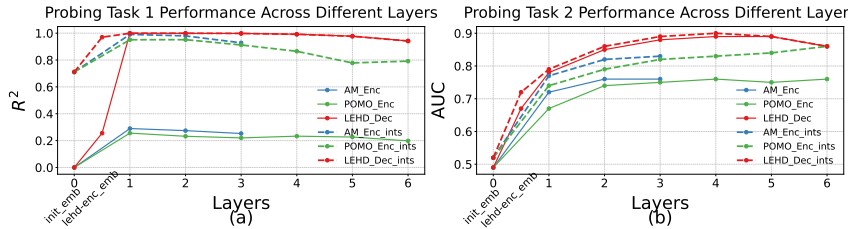

Figure 2: Probing results across different layers.

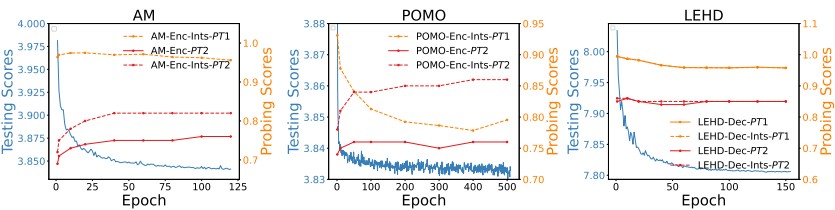

Figure 3: Probing results during NCO models training.

right from the start of training, indicating that the LEHD model has already learned how to avoid myopic decisions early in the process. What additional information LEHD learns to make its node selection decisions could be further investigated in future research by designing new probing tasks.

**Do NCO embeddings encode constraint information, as demonstrated with the CVRP?**

We address this by investigating *Probing Task 3* and *Probing Task 4*, which target a critical constraint in CVRP: the capacity constraint. The former focuses on a low-level feature, the linear additive relationship among node demands, while *Probing Task 4* explores a high-level aspect, examining whether the embeddings produced by NCO models encode information about whether nodes belong to the same route in the optimal solution. Through probing, we verify that NCO models are indeed capable of capturing such knowledge. Detailed results and analysis can be found in Appendix C.1.2.

### 3.3 Probing for Border NCO

**How robust is probing when used to explore NCO models?**

We further investigate the robustness of probing in analyzing NCO models through three additional studies: (1) Besides transformer-based NCO models, can probing also be applied to other types of models? To examine this, we conduct a preliminary experiment on DIFUSCO [18], a diffusion-based NCO model in Appendix C.2.1. The results confirm that the embeddings of diffusion-based models can also be effectively analyzed using probing. More in-depth investigations, or explorations of additional diffusion-based NCO models [19, 20], would be valuable to further illuminate their underlying mechanisms. (2) Can probing be used to explore information in non-Euclidean spaces? In Appendix C.2.2, we examine this question using the Asymmetric Traveling Salesman Problem (ATSP) as a case study, and demonstrate that probing is indeed applicable in non-Euclidean settings. (3) Can probing be used to analyze other combinatorial optimization problems, or applied to architectures beyond attention-based NCO models? In Appendix C.2.3, we address these questions by applying probing to the Job Shop Scheduling Problem (JSSP) and a corresponding GNN-based neural model. We use probing to investigate whether the embeddings of NCO models can capture precedence constraint information. The results confirm that probing remains effective in interpreting other deep learning architectures and combinatorial optimization tasks.

**Can probing be used to investigate the performance differences among NCO models?**

In addition to revealing what types of knowledge are captured by the embeddings of NCO models, the traditional probing methods also provide indirect visions into why different NCO models exhibit varying performance. For example, by analyzing probing performance across problem sizes, we observe that LEHD achieves better probing results on larger instances (200-TSP), which aligns with its

stronger performance in large scale TSP, thereby offering supporting evidence from a representational perspective. Additionally, we perform ablation studies on LEHD to investigate the impact of different components and the performance when embedding different nodes, which further strengthens the structural design claims of LEHD. For detailed analysis, please refer to Appendix C.3.

However, a more compelling question is whether probing can reveal the internal mechanisms of black-box DL models, thereby providing direct—rather than indirect—evidence to explain performance differences and uncover the factors contributing to generalization. In the following Section 4, we demonstrate how our proposed CS-Probing method can effectively provide such direct evidence.

# 4 Opening the Black Box of NCO Models Using CS-Probing

In this section, we demonstrate how our proposed CS-Probing method leads to three key findings: (1) it reveals the distinct inductive biases learned by different NCO models; (2) it uncovers the differing generalization mechanisms across models; and (3) it identifies and localizes key embedding dimensions that encode task-specific knowledge. Based on these findings, we further demonstrate the practical value of probing by validating how the analyzed model can enhance its generalization through insights derived from CS-Probing.

## 4.1 CS-Probing: A New Tool

In addition to systematically designing two sets of high-level and low-level probing tasks tailored for the combinatorial optimization problem, we also propose a novel probing analysis tool: analyzing both the absolute magnitude and of the coefficients in a linear probing model. We refer to this method as *Coefficient Significance Probing* (CS-Probing). Our proposed CS-Probing enables a more fine-grained analysis by examining the role of each individual embedding neuron (or dimension) in capturing specific knowledge. In this section, we use the first two TSP-related probing tasks (i.e., *Probing Task 1* and *Probing Task 2*) as examples to demonstrate how CS-Probing analyzes NCO models. Other probing tasks can also be analyzed using CS-Probing, and the corresponding results demonstrating that NCO models excel at capturing simple additive constraint-related information when solving CVRP are presented in Appendix D.5.

## 4.2 Inductive Biases

We began by examining the embeddings from different NCO models just before the decision output layer and observed clear differences in their activation patterns. Figure 4 presents a heatmap of node embeddings across the NCO models. Specifically, fifty instances are sampled from each of the three NCO model datasets, and the embedding of a specific node from each instance is visualized. The results show that LEHD exhibits strong activation concentrated in fewer than 20 fixed dimensions, with absolute values often in the tens. In contrast, AM and POMO exhibit different inductive biases, characterized by more dispersed activation patterns, with no consistently dominant dimensions and significantly smaller coefficient magnitudes (all below 4 in absolute value).

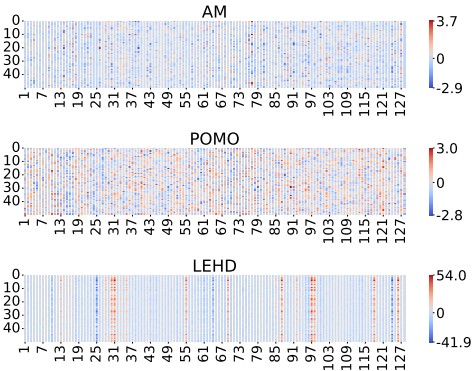

Figure 4: Heatmaps of node embeddings. The x-axis represents different embedding dimensions, and the y-axis represents the instances.

We further investigate the distinct inductive biases learned by the three NCO models through probing, specifically by analyzing the coefficients of the probing models—i.e., via our proposed CS-Probing method. This analysis reveals how the embeddings from different NCO models encode information differently, reflecting their respective inductive biases. In doing so, we demonstrate **how CS-Probing helps uncover the potential reasons behind the superior performance of better-performing NCO models, validating the claim made in the original LEHD paper** [13] that "such

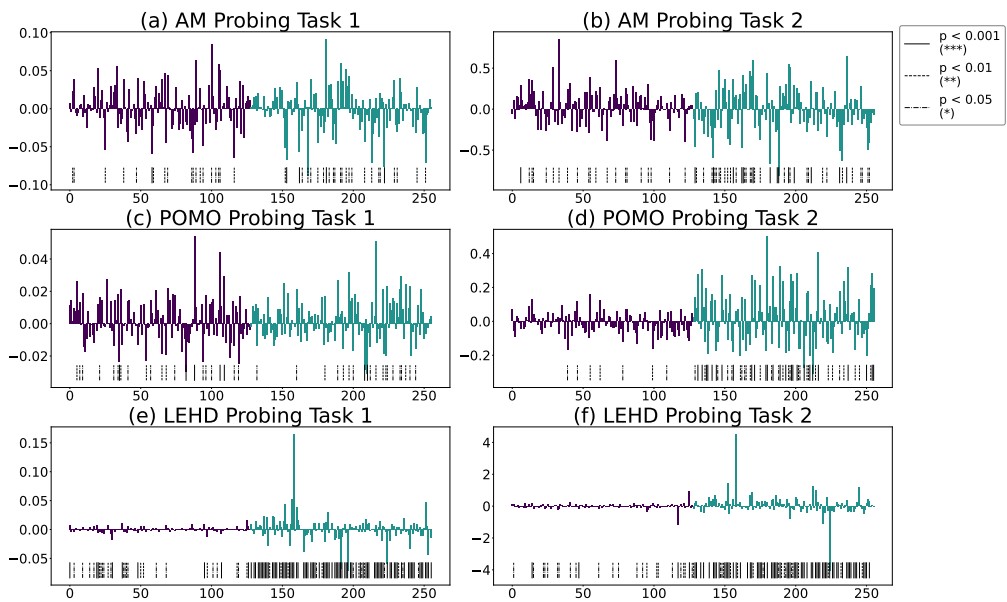

Figure 5: Coefficients of probing models for two TSP-related probing tasks across all NCO models.

a dynamic learning strategy enables the model to adjust and refine its captured relationships between the starting/destination and available nodes".

Figure 5 presents the probing model coefficients obtained on the test sets for *Probing Task 1* and *Probing Task 2* across the three NCO models. In each subplot, the x-axis corresponds to latent feature dimensions, while the y-axis shows the coefficient values. The top portion of each subplot displays the raw coefficient magnitudes, and the bottom portion illustrates the statistical significance of each feature. The first 128 dimensions represent the embedding of the current node, which is the node being visited during the autoregressive decision process of the NCO model, and the next 128 dimensions correspond to the embedding of a candidate node, that is, an unvisited node.

From Figure 5, we observe that LEHD, the best-performing model among these three NCO models, exhibits more statistically significant features in its node embeddings for both TSP-related probing tasks compared to AM and POMO. Specifically, examining the coefficients of each node's embeddings of LEHD reveals that for the current node, the probing model's coefficients tend to have smaller absolute values, with only a few being statistically significant. In contrast, the embeddings of candidate nodes (those relevant to the decision-making process for selecting the next node to visit in the current step) have a greater number of statistically significant dimensions. Additionally, we confirm that during training, LEHD progressively develops this property, gradually learning to encode knowledge into specific embedding dimensions, as illustrated in Figure 14 in Appendix D.1.

If we disregard the absolute values of the coefficients and focus solely on statistical significance, this pattern is also observed in AM and POMO during the myopia-avoidance task, albeit with far fewer statistically significant features in their embeddings compared to LEHD. However, when it comes to perceiving Euclidean distances, the embeddings of AM and POMO as features show no such distinction. In subplots (a) and (c), the coefficients and the number of significant features for the two nodes' embeddings are similar, regardless of their roles as the current node or other nodes.

## 4.3 Generalization Mechanisms

We extend the analysis to explore **how CS-Probing can uncover direct evidence of generalization in NCO models**. Specifically, we show that NCO models with superior generalization performance learn transferable representations, that is, features that generalize beyond the training distribution and enable robust performance on unseen problem instances.

Table 2 shows the CS-Probing results of AM, POMO, and LEHD on two TSP probing tasks, specifically the top 5 dimensions with the largest absolute coefficients. The numbers indicate the dimension indices (starting from 1), with parentheses indicating whether the dimension comes from the current node or the candidate node. Bold entries highlight dimensions that are reused during

generalization, and underlined entries further indicate that their relative ranking remains consistent. To further ensure reliability, we perform multiple hypothesis testing. Specifically, we apply the Benjamini-Hochberg procedure to control the false discovery rate (FDR) at 0.05 across the 256 tested dimensions. The key LEHD dimensions identified earlier remain significant after correction, consistent with the results in Table 2.

The results reveal that in less generalizable models (AM and POMO), the key embedding dimensions associated with the probing tasks vary across generalization scenarios. In contrast, the more generalizable model (LEHD) consistently maintains the same top-2 dims, i.e., the 31th and 69th for *Probing Task 1*, and the 31th and 97th for *Probing Task 2*. Figure 15 (i)-(l) in Appendix D.2 visualizes this result for the LEHD model, intuitively demonstrating how it consistently utilizes fixed dimensions to capture relevant information during generalization. In contrast, the visualizations for AM and POMO are presented in Figures 15 (a)-(d) and 15 (e)-(h), respectively.

These findings collectively support the conclusion that CS-

Table 2: Top 5 dimensions from CS-Probing results two TSP probing tasks on TSP-20 and TSP-100 across three models.

|  |  | Top_n | *20 train / 20 test* | | | *20 train / 100 test* | | |
| --- | --- | --- | --- | --- | --- | --- | --- | --- |
|  |  |  | Dim. | Coef. | Sig. level | Dim. | Coef. | Sig. level |
| Probing Task 1 | AM | 1 | 54 (candidate node) | 0.0917 | *** | 86 (candidate node) | -0.1031 | ** |
|  |  | 2 | 41 (candidate node) | -0.0884 | ** | 34 (current node) | 0.1015 |  |
|  |  | 3 | 101 (current node) | -0.0848 | * | 22 (candidate node) | 0.0770 | * |
|  |  | 4 | 95 (candidate node) | -0.0767 | *** | 54 (current node) | 0.0746 | * |
|  |  | 5 | 124 (candidate node) | -0.0712 | ** | 21 (candidate node) | 0.0740 |  |
|  | POMO | 1 | **89 (current node)** | 0.0539 | *** | **89 (candidate node)** | 0.0562 | * |
|  |  | 2 | **89 (candidate node)** | 0.0510 | * | 80 (current node) | -0.0404 | ** |
|  |  | 3 | 107 (current node) | 0.0443 | *** | **89 (current node)** | 0.0369 |  |
|  |  | 4 | 70 (candidate node) | 0.0322 | ** | 104 (current node) | 0.0359 |  |
|  |  | 5 | 83 (candidate node) | -0.0308 | *** | 24 (candidate node) | 0.0349 |  |
|  | LEHD | 1 | **31 (candidate node)** | 0.1651 | *** | **31 (candidate node)** | 0.1703 | *** |
|  |  | 2 | **69 (candidate node)** | -0.0718 | *** | **69 (candidate node)** | -0.0793 | *** |
|  |  | 3 | 64 (candidate node) | -0.0683 | *** | 118 (candidate node) | 0.0708 | *** |
|  |  | 4 | 97 (candidate node) | -0.0604 | *** | 25 (candidate node) | 0.0630 | *** |
|  |  | 5 | 30 (candidate node) | 0.0526 | *** | 98 (candidate node) | 0.0509 | *** |
| Probing Task 2 | AM | 1 | 34 (current node) | 0.8561 | ** | 41 (candidate node) | 1.6827 | *** |
|  |  | 2 | 61 (candidate node) | -0.8153 | *** | 34 (candidate node) | 1.3354 | ** |
|  |  | 3 | 55 (candidate node) | -0.6612 | *** | **106 (candidate node)** | 1.1430 | ** |
|  |  | 4 | 109 (candidate node) | 0.6406 | *** | 70 (candidate node) | -0.9718 | ** |
|  |  | 5 | **106 (candidate node)** | -0.6329 | ** | 54 (candidate node) | 0.9681 | *** |
|  | POMO | 1 | 53 (candidate node) | 0.5004 | *** | 125 (candidate node) | 0.7499 | *** |
|  |  | 2 | 89 (candidate node) | 0.4088 | *** | 86 (candidate node) | -0.6989 | *** |
|  |  | 3 | 62 (candidate node) | 0.3238 | *** | 70 (candidate node) | 0.5580 | *** |
|  |  | 4 | 110 (candidate node) | 0.3217 | *** | 121 (candidate node) | -0.4910 | * |
|  |  | 5 | 7 (candidate node) | 0.3081 | *** | 82 (candidate node) | 0.4776 | *** |
|  | LEHD | 1 | **31 (candidate node)** | 4.5238 | *** | **31 (candidate node)** | 2.7083 | *** |
|  |  | 2 | **97 (candidate node)** | -4.0741 | *** | **97 (candidate node)** | -1.6566 | *** |
|  |  | 3 | 25 (candidate node) | 1.4400 | *** | 126 (candidate node) | 1.3297 | *** |
|  |  | 4 | 85 (candidate node) | 1.2421 | *** | 30 (candidate node) | -1.2525 | *** |
|  |  | 5 | 118 (current node) | -1.1680 | *** | 42 (candidate node) | 0.7862 | *** |

Probing explains the generalization behavior of NCO models by revealing that they consistently reuse the same embedding dimensions to encode specific knowledge (as observed in LEHD). Besides, the results indicate that when the knowledge encoded in specific dimensions becomes disorganized during generalization, model performance deteriorates. To further support this argument, we examine whether AM and POMO, trained on 20-node instances, can generalize not to the distant 100-node setting, but to instances that are closer in scale (e.g, with 21, 25, or 30 nodes). The results show that both models exhibit similar reuse of embedding dimensions when generalizing to 21-node instances, suggesting that when generalization is achievable, NCO models tend to capture specific knowledge using a fixed (and small) set of dimensions. The results are listed in Table 14 in Appendix D.3.

## 4.4 Key Embedding Dimensions

Previously, We identify key factors influencing NCO model generalization and provide direct evidence of their performance. Notably, the LEHD model consistently reuses the same top-2 embedding dimensions across different probing tasks and problem scales. Building on this insight, we further demonstrate **the practical value of CS-Probing** from multiple perspectives.

First, the two key dimensions identified by CS-Probing offer interpretability into how NCO models make decisions within a high-dimensional latent space that is otherwise difficult for humans to comprehend. For example, we examine the LEHD model's behavior on *Probing Task 2* (avoiding myopic decisions) by the 2D plane formed by the two key dimensions of LEHD's embedding (dimensions 31 and 97). Figure 6 illustrates the result for one instance with the random seed set to one. The left plot shows the optimal solution, and the middle shows the greedy solution. The right plot visualizes the values of the two key dimensions from the final node embeddings output by LEHD (before the softmax layer). In this case, the current node is node 4. A myopic decision based on Euclidean distance would choose node 5, whereas the optimal solution selects node 3. In the 2D space defined by the identified key dimensions, node 4 is indeed closer to node 3 than to node 5. Furthermore, other nodes that are closer to node 4 in this space also appear nearby in the optimal tour, further supporting our interpretation.

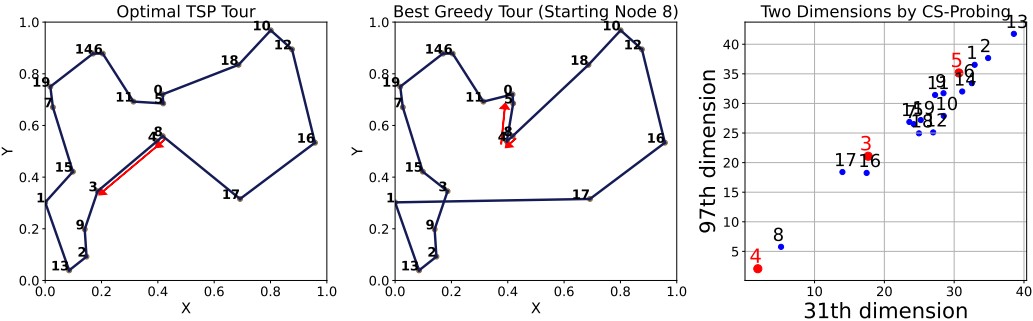

Figure 6: Routing solutions (see Appendix B.1.2 for details on how to obtain) and the key two-dimensional embeddings in the node representation of LEHD.

These results reveal how black-box models make decisions by identifying the most informative embedding dimensions that encode the knowledge required for specific decision types. Additional results on other random seeds are in Appendix D.4.

The second practical value of CS-Probing lies in its ability to validate model architecture and provide insights for future model design. Regarding the relationship between CS-Probing findings and model structure, we observe from the 2D plots of the two key dimensions (Figures 6, 16) that LEHD not only avoids myopic behavior but also consistently positions the current node in the bottom-left corner, clearly separated from other nodes. This spatial separation directly reflects the effect of LEHD's architectural design, in which candidate node embeddings are re-encoded at each decision step to enhance the capture of decision-relevant information.

Table 3: Results of only two dimensions from LEHD's output embedding.

| Problem | Dimension(s) | Optimality Gap |
|---|---|---|
| 100 | All 128 | 0.57% |
| | 31, 97 | 0.65% |
| | 32, 97 | 0.75% |
| | 21, 123 | 183.80% |
| | 7, 78 | 384.19% |
| 200 | All 128 | 0.86% |
| | 31, 97 | 0.93% |
| 500 | All 128 | 1.56% |
| | 31, 97 | 1.81% |
| 1000 | All 128 | 3.17% |
| | 31, 97 | 3.56% |

Table 4: Performance degradation from zeroing key vs. non-key dimensions in LEHD.

| Zeroing Dimension(s) | Optimality Gap |
|---|---|
| None(original) | 0.57% |
| 31, 97 | 60.43% |
| 31 | 0.90% |
| 97 | 1.80% |
| 98 | 0.59% |
| 126 | 0.60% |
| 98, 126 | 0.62% |

Thirdly, across multiple results and insights obtained by CS-Probing above, we find that LEHD's final-layer embeddings, just before the output layer, can capture both low-level and high-level knowledge using only a small number of dimensions. To explore whether the model can make decisions using only these key dimensions, we conduct an experiment where we retain only the two most important dimensions (as identified by CS-Probing) from LEHD's 128-dimensional output. Remarkably, the model still achieves nearly equivalent performance. In particular, we identify two key dimensions of a LEHD model trained on 100-node instances using CS-Probing, as described in the previous sections. We then evaluate LEHD models trained on instances of other scales using only these two dimensions, and they continue to generalize effectively to 200-, 500-, and even 1000-node problems, yielding results comparable to those obtained using the full embedding. See Table 3 for detailed results. To further validate, we conduct neuron ablation: zeroing the two key dimensions (31 and 97) causes performance to collapse (more than 60% gap), while zeroing non-key but seemingly important dimensions yields little impact (Table 4). Specifically, dimension 98 is chosen because it exhibits large activation values in the embedding visualization (from Figure 4), and dimension 126 is selected because it has the highest probing coefficient among all non-key dimensions (from Table 2).

One key insight emerging from this finding is the potential benefit of imposing regularization or constraints on embedding dimensionality during LEHD model training. This raises important questions about whether dimensional efficiency could be further improved through its architectural or training adjustments. In the following section, we conduct a preliminary exploration of this idea. For

future work, it would be valuable to explore how embedding dimensionality should be configured across different layers and training stages to enhance both efficiency and generalization.

## 4.5 Practical Implications

We conduct a simple yet effective experiment based on insights from CS-Probing. Since LEHD generalizes better than other models by relying on a small subset of embedding dimensions, we introduce a regularization term to promote sparsity in its final-layer embeddings. With only minor code changes, this improves generalization. As shown in Table 5, we train LEHD with different regularization strengths ($\lambda$) on TSP100 and evaluate transfer to TSP200 and TSP1000, measuring the gap to best known solutions. The baseline (origin, namely $\lambda$=0) corresponds to the unregularized model, while moderate regularization ($\lambda$=1e-6 to 1e-3) improves generalization at larger scales. We also test cross-distribution generalization by training on uniform instances and testing on TSPLib. These results provide preliminary evidence that regularization may enhance generalization performance and highlight the potential of probing insights as an analytical tool for informing model design.

Table 5: Results of LEHD models trained with different regularization strengths ($\lambda$), demonstrating generalization performance across problem scales and distributions.

|  | origin | 1e-6 | 1e-5 | 1e-4 | 1e-3 |
|---|---|---|---|---|---|
| TSP100 | 0.57% | 0.58% | 0.57% | 0.57% | 0.57% |
| TSP200 (generalization) | 0.86% | 0.88% | 0.86% | **0.73%** | 0.82% |
| TSP1000 (generalization) | 3.17% | **2.87%** | 2.93% | 2.97% | 3.05% |
| TSPLib(generalization) | 5.26% | **4.94%** | 4.99% | 5.05% | 8.61% |

It is important to note that these analyses are not intended to argue that a particular transformer-based architecture is inherently superior, nor that all models should employ sparsity-inducing regularization. Rather, by using LEHD as a case study, we aim to validate the practical value of CS-Probing. Overall, this work presents the first systematic attempt to apply probing to NCO research, introducing the CS-Probing tool as a means of analyzing and better understanding the internal mechanisms of NCO models. This opens up a new pathway for both analyzing and improving NCO models. We believe that extending these analyses to a broader set of models in the future will enable more transparent and trustworthy pathways for advancing research on NCO models.

## 5 Conclusion

In this paper, we introduce *probing* into the study of NCO models to systematically investigate their learned embeddings and to advance the understanding of the internal mechanisms underlying these black-box approaches. We systematically design probing tasks and employ our proposed CS-Probing method to investigate what decision-related knowledge NCO models capture and how they encode this knowledge. Additionally, we demonstrate the practical value of probing by providing empirical support for claims regarding the design of NCO models, offering evidence to explain mechanisms such as generalization performance, and generating insights to guide future research in the NCO field.

One of these insights is that inference can achieve comparable results using only two key dimensions discovered through CS-Probing. This finding highlights a potential avenue for compressing, distilling, and pruning the representation space of NCO models, as well as investigating the dimensionality of NCO model representations.

Another potential direction is to explore additional knowledge to further enhance the transparency and interpretability of NCO models. Much like how DNA sequencing revolutionized genetics, our proposed CS-Probing offers a powerful lens for understanding NCO models by identifying specific dimensions within high-dimensional embeddings that encapsulate essential knowledge. As more probing tasks emerge, this approach has the potential to progressively transform black-box representations into interpretable, structured forms, offering a promising and impactful direction for the field. Enhancing transparency and interpretability could significantly promote the broader application of deep learning models in scientific and engineering domains.

## Acknowledgments

This research is supported by the National Research Foundation, Singapore under its AI Singapore AI Research Fundamental Research Collaborative (US-NSF Researcher Call) (AISG Award No: AISG3-RP-2025-036-USNSF). This research is also supported by the National Research Foundation, Singapore under its AI Singapore Programme (AISG Award No: AISG2-100E-2023-118). Any opinions, findings and conclusions or recommendations expressed in this material are those of the author(s) and do not reflect the views of National Research Foundation, Singapore.

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

# A Related Work

## A.1 NCO

Neural (deep learning-based) methods have been applied to various combinatorial optimization problems for several years [21, 22]. With the rapid advancement of deep learning (DL), an increasing number of approaches have been introduced to address these classical problems in operations research (OR). In the context of the routing problem discussed in this paper, researchers have explored methods such as graph convolutional networks (GCNs) [23, 24], pointer networks with recurrent neural networks (RNNs) [25, 26], diffusion-based approaches [18, 19, 20], and attention mechanisms [11, 27, 12, 28, 29, 30, 13], which are the primary focus of this study.

### A.1.1 AM, POMO, LEHD

AM [3] [11] is one of the earliest and most successful attention mechanisms-based models for routing tasks. It is pioneering in introducing the widely popular Transformer architecture to combinatorial optimization problems, inspiring a multitude of subsequent models. POMO [4] [12], as a notable example, retains a structure fundamentally similar to AM (with minor differences, such as in context embedding) but introduces a novel reinforcement learning (RL) training method.

AM not only introduces the Transformer architecture but also makes significant contributions to the model design for routing problems. A notable idea is AM's context embedding in the decoder, which focuses on the current node and the starting node (for TSP problems). Although many later models do not adopt this exact context embedding design, the core idea of focusing on these two key nodes remains. For example, even though LEHD's decoder design differs from AM's, it fundamentally considers how to represent information from these two critical nodes.

Specifically, the difference between LEHD [5] [13] and AM lies in their architectural design. Figure 1 illustrates the architecture of both models. In Figure 1(a), AM uses a multi-layer encoder to learn how to represent node information based on their input features (coordinates), while the decoder performs a single attention computation on the node representations generated by the encoder, producing a global "glimpse" for decision-making without updating the node embeddings. This is known as the "Heavy Encoder Light Decoder" structure. In contrast, LEHD adopts a "Light Encoder Heavy Decoder" structure, where the encoder uses only a single attention layer to learn node representations, while the decoder, at each step, re-learns the embeddings of the current node, destination node, and candidate nodes through multiple attention layers. In LEHD, as shown in Figure 1(b), $h_s$ and $h_d$ represent the embeddings of the current node (referred to as the starting node in LEHD) and the destination node, while the node embeddings located in the middle are filtered in LEHD to exclude previously visited nodes.

### A.1.2 Concerns

Although NCO methods have seen rapid development in academia, industries remain cautious about deploying them to replace classical OR methods. This is because these DL-based methods are perceived as black-box models, lacking the reliability and interpretability of traditional OR approaches. As a result, even though some NCO models have achieved strong performance on certain instances, they are still met with skepticism. For example, [31] raises concerns about the overuse of GNNs, noting that the improvements achieved by GNN-based methods over traditional distance-related approaches were minimal. To address this, we are the first to unveil the inner workings of NCO models, aiming to enhance understanding of their internal mechanisms.

## A.2 Probing

The probing method used in this paper was initially applied to understand the representations of DL models in computer vision [5] and natural language processing [6, 14, 17, 15, 32]. Beyond traditional DL tasks, probing has also demonstrated effectiveness in other domains, such as exploring world representations [33, 10] and in-context learning [9] in large language models, auditory representations

---

[3] `https://github.com/wouterkool/attention-learn-to-route`
[4] `https://github.com/yd-kwon/POMO/tree/master`
[5] `https://github.com/CIAM-Group/NCO_code/tree/main/single_objective/LEHD`

[34, 35], and studying the quality of unsupervised reinforcement learning representations [36]. A systematic use of probing to analyze a deep model's ability to extract and store knowledge can be seen in [37], which investigates how large language models encode vast amounts of world knowledge as a case study.

In the field of NLP, prior work such as [15, 16] has explored the use of probing techniques to identify key neurons, aiming to analyze the knowledge learned by deep learning models. Notably, [16] demonstrated that the learned patterns in the feed-forward layers of Transformer models are human-interpretable: lower layers tend to encode shallow syntactic features, while higher layers capture more abstract semantic representations. Similarly focusing on neurons in feed-forward layers, [10] identified individual "space neurons" and "time neurons" that consistently encode spatial and temporal coordinates in large language models, by computing the cosine similarity between each neuron's input or output weight vector and a predefined probe direction vector. In contrast to these studies, our work focuses on NCO models and investigates how they represent problem-specific input information—such as nodes in routing problems. Accordingly, we primarily analyze the model's embeddings, where "neurons" in our context refer to individual dimensions within the embedding space. While our current analysis centers on the embedding layer, we believe that probing the feed-forward layers of Transformer-based NCO models presents a promising direction for future research.

# B   Design and Setup of Probing Tasks

The steps for using probing to explore deep learning representations are as follows: first, define the probing task based on the target information to be explored; second, collect the labels required for the probing dataset, ensuring they are relevant to the target information; third, combine the embeddings from the deep learning model with the labels to complete the probing dataset; and finally, train the probe and evaluate its performance on out-of-sample data. Figure 7 shows the process of creating and labeling the probing dataset.

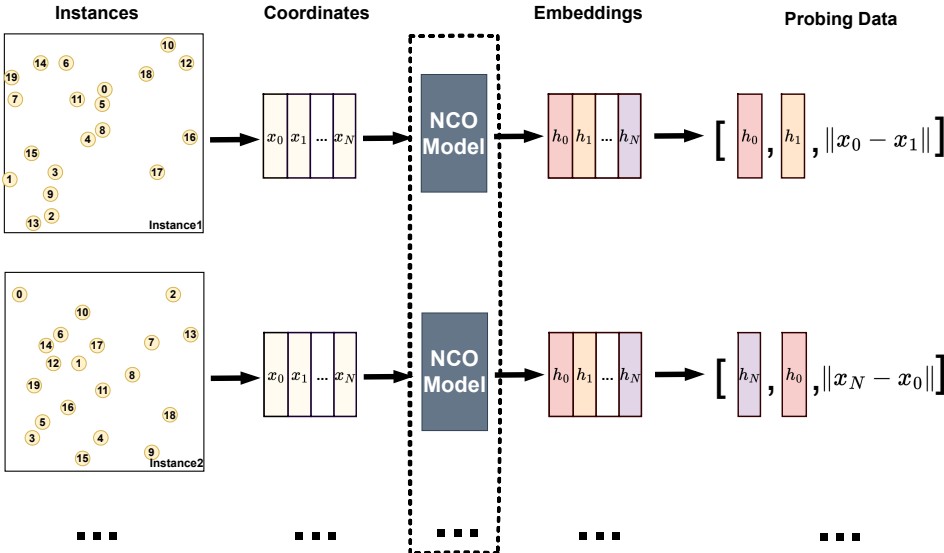

Figure 7: The process of creating the dataset for *Probing Task 1* is illustrated from left to right. For a given instance, we input its complete data (all nodes) into the NCO model being probed (with the dashed box representing the same NCO model). We then extract the embeddings from the probed part (e.g., the encoder or decoder) or layer of the model and select the corresponding embeddings of the required nodes as features. These features, combined with the label, form a single data point.

### B.1 Four Probing Tasks

#### B.1.1 *Probing Task 1*: Euclidean Distance

When solving routing problems in Euclidean space, the Euclidean distance between nodes is a critical piece of information for all solution methods. For instance, a simple greedy algorithm for solving the TSP starts at an arbitrary node, computes the Euclidean distance between the current node and all unvisited nodes, and selects the nearest one as the next destination. This process is repeated until all nodes are visited, returning to the starting node to form a Hamiltonian cycle. In traditional methods, whether using exact approaches (mathematical programming) that rely on the distance matrix of nodes as input or approximate (heuristic) methods [38, 39], the Euclidean distance between any two nodes must be precomputed or computed on the fly. Therefore, for a TSP solver, recognizing the Euclidean distances between nodes is essential. Based on this, we aim to explore whether a trained learning-based NCO model can capture this critical Euclidean distance between the current node and any of the candidate nodes in its representations.

**Probing task.** *Probing Task 1* aims to examine whether the embeddings of NCO models encode the distance between the current node and any of the candidate nodes during decision-making. Given the embeddings of two nodes, a probing model is trained to directly predict the Euclidean distance between them. This probing task, which takes two embeddings as input features, is similar to the probing tasks used in NLP to evaluate pairwise relations between words [8].

**Dataset.** Figure 7 illustrates the process of creating a sample for *Probing Task 1* and its corresponding dataset. Given the current node $n_i$ and any randomly selected node $n_j$ from the candidate nodes, we extract their embeddings $h_i$ and $h_j$ from the relevant layers of the NCO model we want to probe. The embeddings of the two nodes are then concatenated into a feature vector $[h_i, h_j]$, with the Euclidean distance between $n_i$ and $n_j$ serving as the label. By collecting sufficient data in this manner, we construct the dataset for *Probing Task 1*. Since the label (i.e., the distance) is a continuous, *Probing Task 1* is framed as a regression prediction task.

#### B.1.2 *Probing Task 2*: Avoidance of Myopia

Selecting the next unvisited node solely based on the nearest Euclidean distance, as in the greedy algorithm, will not result in the optimal solution from a global perspective. This approach is often described as "myopic", and many efforts have been made to avoid such shortsighted strategies [40, 41, 42, 43]. A well-designed NCO model must similarly learn to avoid myopic strategies and adopt a more global perspective to solve the problem effectively. To investigate this, we design *Probing Task 2* to explore whether the embeddings of NCO models exhibit the ability to avoid shortsighted decisions at a given step.

**Probing task.** We define *Probing Task 2* as a binary classification task, where the probing model is trained to determine whether the current node (e.g., $n_i$) should be linked to node $n_j$. Node $n_j$ could either be a myopic choice that leads to a local optimum or the node connected to $n_i$ in the global optimal solution. To assess whether the NCO models make myopic decisions by choosing the nearest Euclidean distance, we construct data points as illustrated in Figure 8.

**Dataset.** First, we randomly generate an instance with $N$ nodes, input it into a mathematical programming model, and use the Gurobi [44] solver to obtain the theoretical optimal solution, as shown in Figure 8(a). Next, starting from each node, we use a greedy algorithm to generate $N$ solutions and select the best one (as illustrated in Figure 8(b), gradually comparing the next node selected by the greedy algorithm with the optimal solution. For example, in the instance shown in Figure 8, when the current node is node 4, the optimal solution selects node 3, whereas the greedy algorithm selects the nearest one, node 5. Ultimately, we obtain two data points for this instance: node 4 connected to node 3 represents the optimal choice, labeled as a positive example (i.e., the feature is $[h_4, h_3]$ and the label is 1), while node 4 connected to node 5 represents the myopic choice of the greedy algorithm, labeled as a negative example (i.e., the feature is $[h_4, h_5]$ and the label is 0).

**Domain knowledge.** Unlike the relatively straightforward probing tasks and datasets in CV and NLP, probing in the CO field requires incorporating domain-specific knowledge. For instance, in this dataset, there may be multiple optimal solutions. Suppose one of them includes node 4 connected to node 5, which would render a label of 0 incorrect. To verify this, we add a constraint to the mathematical model that forces the connection between nodes 4 and 5. The new optimal solution

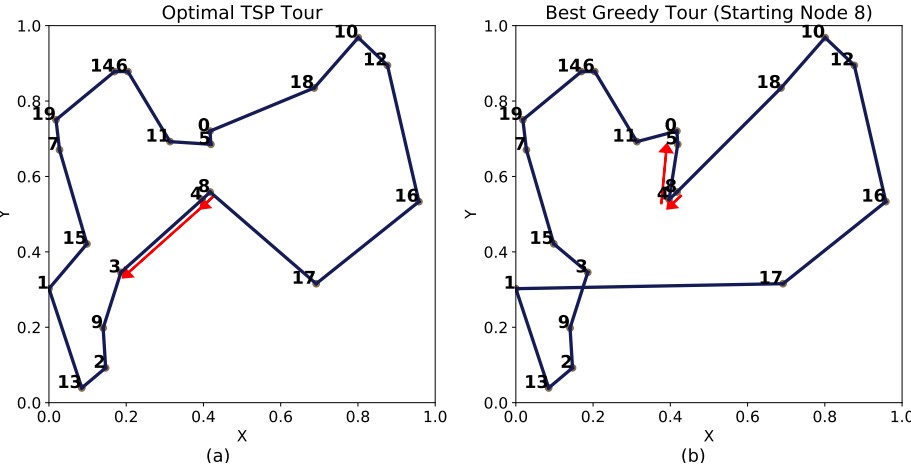

Figure 8: An example of solutions to the TSP for a specific instance: (a) represents the optimal solution generated by the mathematical model and solved using Gurobi; (b) shows the best solution obtained through a greedy algorithm.

obtained under this constraint is worse than the original solution without the constraint. Similarly, for data labeled 1, we add a constraint preventing the connection between nodes 4 and 3, and the resulting solution is also worse. This confirms that both labels are valid.

### B.1.3 *Probing Task 3*: **Perception of Constraints**

For the TSP problem, the first two probing tasks provide a comprehensive analysis of the representational capacity of NCO models. However, for more complex VRP, where additional constraints are introduced, we are curious whether NCO models can capture these constraint-related information. If not, it suggests that NCO models might merely rely on masking to artificially limit their outputs. This would imply an inherent limitation in how NCO models handle constraints.

**Probing task.** To answer this question, we design *Probing Task 3* to explore whether NCO models can capture the knowledge required to determine the feasibility of the capacity constraint in the CVRP problem. Since the capacity constraint primarily involves the linear (additive) relationship among the demands of nodes, we design *Probing Task 3* to check whether the embeddings of two nodes can represent the sum of their demands. Thus, for *Probing Task 3*, a probing model is trained to predict the sum of the demands given the embeddings of two nodes.

**Dataset.** We extract the embeddings of two nodes, $h_i$ and $h_j$, from the relevant layers of the NCO model being probed. Unlike the previous non-linear probing tasks, predicting the sum of two demands—a linear addition task—may be inherently too simple. Therefore, a linear probing model might not be sufficient to demonstrate whether the NCO model can capture this knowledge. To delve deeper, in addition to concatenating the embeddings of the two nodes ($[h_i, h_j]$) as the input for the probing task, we also apply Hadamard product on the two embeddings, $[h_i \odot h_j]$, as an alternative input. The latter approach aims to simulate the attention computation process in attention-based NCO models (as in most models where the decoder ultimately uses attention to compute a compatibility score to determine node selection probability), allowing us to examine whether the model can capture the additive effect of demand features.

### B.1.4 *Probing Task 4*: **Same Route**

*Probing Task 3* investigates a low-level feature related to the capacity constraints in CVRP—whether the embeddings of nodes contain information about their demands. Next, we explore a higher-level, more abstract feature: determining whether two nodes belong to the same route. In the CVRP solution, there are multiple routes, and if two nodes are on the same route in the optimal solution, a solution where they are not on the same route is highly unlikely to be optimal. If NCO models

can perceive this information from a global perspective while solving CVRP, they are more likely to achieve higher-quality solutions closer to the optimal.

**Probing task.** We designed *Probing Task 4* to explore this, formulating it as a binary classification problem. The input to the probe consists of the embeddings of two nodes, while the output is a binary value indicating whether the two nodes belong to the same route.

**Dataset.** After generating the CVRP instances, we use the HGS [4] to obtain approximate optimal solutions due to the large problem size. The other steps are similar to those in *Probing Task 2* and will not be elaborated on here.

## B.2 Summary and Statistics of Four Probing Task Datasets

### B.2.1 Routing Instances and Probing Datasets

For *Probing Task 1* and *Probing Task 2*, we generate 10,000 TSP instances with 20 nodes, 100 nodes, and 200 nodes, respectively, following the method introduced in AM [11], which was subsequently used by both POMO and LEHD. For *Probing Task 3* and *Probing Task 4*, we similarly generate 10,000 CVRP instances with 20 nodes and 10,000 instances with 100 nodes following the method used in AM.

After generating the routing problem instances, we feed them into the NCO model to extract embeddings. Each probing dataset is split into training and test sets, with all reported results based on the test set, i.e., out-of-sample data.

### B.2.2 Overview of Probing Inputs

For a finer-grained analysis, we extract embeddings from different layers and positions, as indicated by the red arrows in Figure 1. Here, we provide a detailed explanation of these extracted embeddings. Another reason for introducing these embeddings is to facilitate the understanding of the main results presented later, specifically the "Probing input" columns in Table 9 and Table 10.

We use the names listed under the "Probing Input" column in these tables to clearly indicate the different embeddings used as inputs. The first segment (AM, POMO, LEHD) indicates from which NCO model the embeddings are extracted. The second segment (Init., Enc., Dec.) represents the different parts of the NCO model from which the embeddings are extracted: initial embeddings, encoder embeddings, and decoder embeddings, respectively.

In the encoder of the NCO model, the initial embeddings (Init.) are extracted before the attention layer, as shown at position P1 in Figure 1. P2 and P5 represent the embeddings from the encoder's attention layers (Enc.), while P6 represents those from the decoder's attention layers (Dec.). We use "$l$" followed by a number to indicate from which specific layer the embeddings are extracted. Specifically, AM's encoder has three layers, POMO's encoder has six layers, and LEHD's encoder has only one layer, while its decoder has six layers.

Since AM and POMO do not update node embeddings in the decoder, their node embeddings in decoder are not included as probing inputs. However, they introduce context embeddings in the decoder to represent the information needed for routing decisions. For example, in solving TSP, the context embeddings are formed by concatenating the embedding of the starting node $h_s$, the current node embedding, and the graph embedding $h_{graph}$—calculated as the mean of all node embeddings. To explore the representational capacity of this design, we also use the context embeddings $[h_i, h_j, h_s, h_{graph}]$ as probing inputs, denoted as "AM-Enc-$l$3-w/c". Additionally, AM uses the context embeddings as a query to compute attention with other node embeddings, generating a glimpse embedding $h_{glimpse}$. To test this, we probe the input $[h_i, h_j, h_{glimpse}]$ and denote it as "AM-Enc-$l$3-w/g". In Figure 1, P3 and P4 represent the positions where the context embeddings and glimpse embeddings are extracted, respectively.

Additionally, for the first two probing tasks, besides using $[h_i, h_j]$ as input, we also consider the element-wise product of the two node embeddings as an interaction term [8], i.e., $[h_i, h_j, h_i \odot h_j]$ as input. Some parts of certain models may linearly combine node embeddings (for instance, many NCO models concatenate the embeddings of nodes and then pass them through a linear projection). In such components of the models, the embeddings are expected to capture decision-relevant information through simple linear combinations. However, embeddings from certain parts in attention-based

models, such as those used to compute a compatibility score among node embeddings through attention mechanisms, may behave differently. In this case, relying solely on the linear input $[h_i, h_j]$ may not fully assess the model's representational capacity. Therefore, we introduce the interaction term $h_i \odot h_j$ to emulate the attention computation. We conduct probing experiments with both input methods: "w/o ints." refers to input without interaction terms $[h_i, h_j]$, and "w/ ints." refers to input with interaction terms $[h_i, h_j, h_i \odot h_j]$, as shown in Table 9 and Table 10.

For *Probing Task 3*, we use both $[h_i, h_j]$ and $[h_i \odot h_j]$ as inputs (the rationale is discussed in the *Probing Task 3* paragraph in Section B.1.3). In Table 10, the "w/o ints." rows correspond to the results for $[h_i, h_j]$, while the "w/ ints." rows correspond to the results for $[h_i \odot h_j]$. Finally, for the 20-node and 100-node instances, we conduct the four probing tasks using NCO models trained on the corresponding scales. The results for both are grouped and presented in Table 9 and Table 10.

Table 6, Table 7, and Table 8 present the specific features, labels, and the number of observations for the different inputs across the four probing tasks.

Table 6: The details of Probing inputs of *Probing task 1*.

| | | Probing input | # Observations | Features | Label |
|---|---|---|---|---|---|
| 20 and 100 | w/o ints. | Coordinates | | $[x_i, x_j]$ | |
| | | AM-Init | | $[h_i, h_j]$ | |
| | | AM-Enc-$l1$ | | $[h_i, h_j]$ | |
| | | AM-Enc-$l3$ | | $[h_i, h_j]$ | |
| | | AM-Enc-$l3$-w/c | | $[h_i, h_j, h_{graph}]$ | |
| | | AM-Enc-$l3$-w/g | 10000 | $[h_i, h_j, h_{glimpse}]$ | $\|x_i - x_j\|$ |
| | | POMO-Enc-$l1$ | | $[h_i, h_j]$ | |
| | | POMO-Enc-$l6$ | | $[h_i, h_j]$ | |
| | | LEHD-Enc-$l1$ | | $[h_i, h_j]$ | |
| | | LEHD-Dec-$l1$ | | $[h_i, h_j]$ | |
| | | LEHD-Dec-$l6$ | | $[h_i, h_j]$ | |
| | w/ ints. | Coordinates | | $[x_i, x_j, x_i \odot x_j]$ | |
| | | AM-Init | | $[h_i, h_j, h_i \odot h_j]$ | |
| | | AM-Enc-$l1$ | | $[h_i, h_j, h_i \odot h_j]$ | |
| | | AM-Enc-$l3$ | | $[h_i, h_j, h_i \odot h_j]$ | |
| | | AM-Enc-$l3$-w/c | | $[h_i, h_j, h_{graph}, h_i \odot h_j]$ | |
| | | AM-Enc-$l3$-w/g | 10000 | $[h_i, h_j, h_{glimpse}, h_i \odot h_j]$ | $\|x_i - x_j\|$ |
| | | POMO-Enc-$l1$ | | $[h_i, h_j, h_i \odot h_j]$ | |
| | | POMO-Enc-$l6$ | | $[h_i, h_j, h_i \odot h_j]$ | |
| | | LEHD-Enc-$l1$ | | $[h_i, h_j, h_i \odot h_j]$ | |
| | | LEHD-Dec-$l1$ | | $[h_i, h_j, h_i \odot h_j]$ | |
| | | LEHD-Dec-$l6$ | | $[h_i, h_j, h_i \odot h_j]$ | |

### B.2.3 Analysis of Input Data

Before conducting each probing task, we begin by analyzing the input probing dataset, using the 20-node dataset as an example for dataset exploration and preprocessing. As this is a regression problem, we first analyze the target variable to observe whether the label distribution is skewed, whether there are outliers, and other characteristics. Figure 9 shows the label distribution for the 20-node dataset in *Probing Task 1* and *Probing Task 3*, with the dataset generation process detailed previously. As seen, the distribution of distances between randomly selected nodes after a current node is chosen approximates a normal distribution. The distribution of demand follows a similar pattern. For *Probing Task 2*, we generate one data point with a label of 1 and one with a label of 0 for each routing instance, resulting in a 1:1 label distribution.

Next, we conducted a feature correlation analysis on the probing dataset. For the probing dataset formed by the embeddings of two nodes (128 dimensions each), there are a total of 256 features. By examining the correlation heatmap in Figure 10, We observe some positive and negative correlations among the 128 dimensions within both single node's embedding, but their number is limited, far fewer than the statistically significant latent features presented in Section 4.2. We can also observe that there are a few scattered stronger correlations in LEHD's embedding, which could be the source of its enhanced ability to retain the perception of Euclidean distance. Additionally, there is no significant correlation between the embeddings of the two nodes.

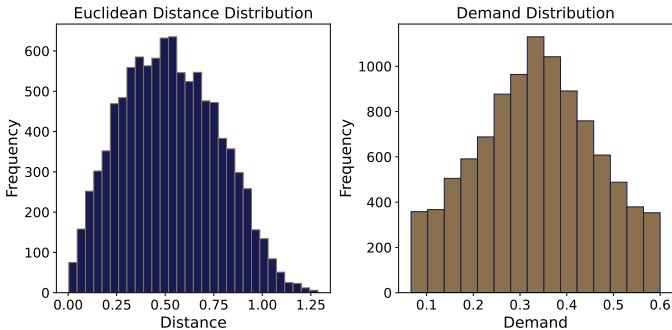

Figure 9: Label distribution for probing datasets in *Probing Task 1* and *Probing Task 3*.

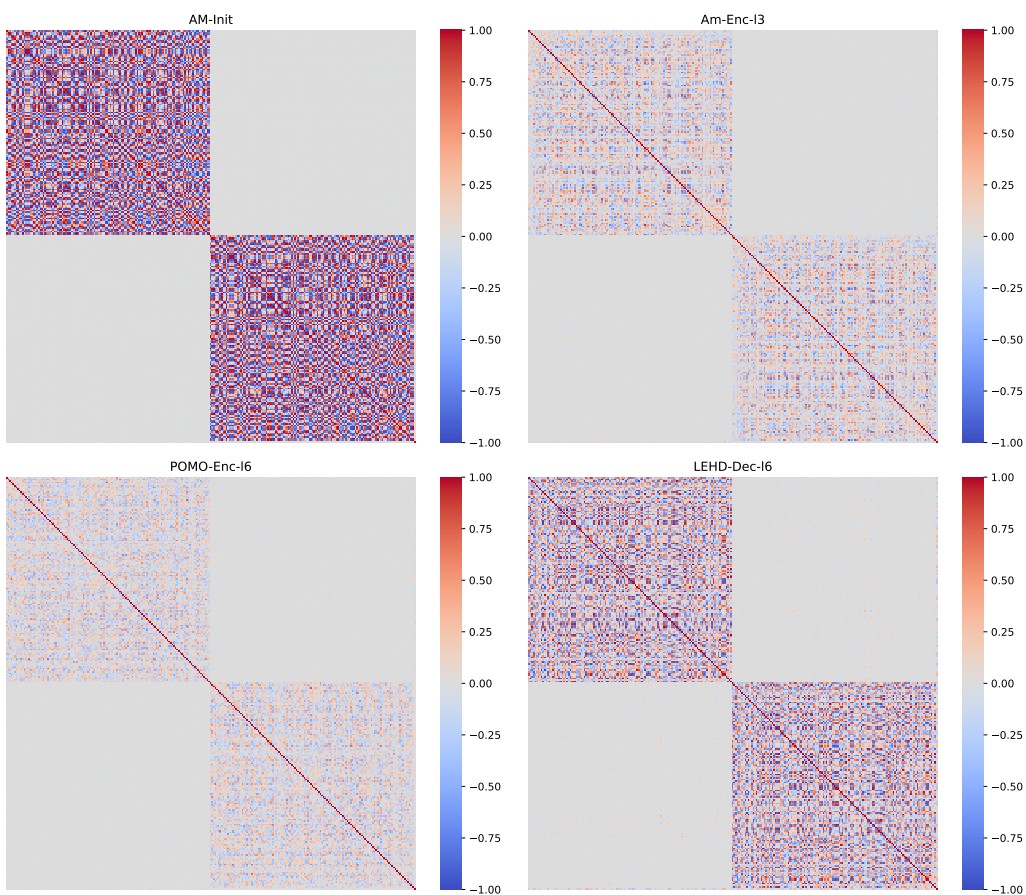

Figure 10: Correlation heatmap for all 256 features (comprising two 128-dimensional node embeddings) of the AM encoder embedding, POMO encoder embedding, and the LEHD decoder embedding.

Table 7: The details of Probing inputs of *Probing task 2* and *Probing task 4*.

| | | Probing input | # Observations | Features | Label |
|---|---|---|---|---|---|
| 20 and 100 | w/o ints. | AM-Init | | $[h_i, h_j]$ | |
| | | AM-Enc-*l*1 | | $[h_i, h_j]$ | |
| | | AM-Enc-*l*3 | | $[h_i, h_j]$ | |
| | | AM-Enc-*l*3-w/c | | $[h_i, h_j, h_{graph}]$ | |
| | | AM-Enc-*l*3-w/g | 20000 | $[h_i, h_j, h_{glimpse}]$ | Binary |
| | | POMO-Enc-*l*1 | | $[h_i, h_j]$ | |
| | | POMO-Enc-*l*6 | | $[h_i, h_j]$ | |
| | | LEHD-Enc-*l*1 | | $[h_i, h_j]$ | |
| | | LEHD-Dec-*l*1 | | $[h_i, h_j]$ | |
| | | LEHD-Dec-*l*6 | | $[h_i, h_j]$ | |
| | w/ ints. | AM-Init | | $[h_i, h_j, h_i \odot h_j]$ | |
| | | AM-Enc-*l*1 | | $[h_i, h_j, h_i \odot h_j]$ | |
| | | AM-Enc-*l*3 | | $[h_i, h_j, h_i \odot h_j]$ | |
| | | AM-Enc-*l*3-w/c | | $[h_i, h_j, h_{graph}, h_i \odot h_j]$ | |
| | | AM-Enc-*l*3-w/g | 20000 | $[h_i, h_j, h_{glimpse}, h_i \odot h_j]$ | Binary |
| | | POMO-Enc-*l*1 | | $[h_i, h_j, h_i \odot h_j]$ | |
| | | POMO-Enc-*l*6 | | $[h_i, h_j, h_i \odot h_j]$ | |
| | | LEHD-Enc-*l*1 | | $[h_i, h_j, h_i \odot h_j]$ | |
| | | LEHD-Dec-*l*1 | | $[h_i, h_j, h_i \odot h_j]$ | |
| | | LEHD-Dec-*l*6 | | $[h_i, h_j, h_i \odot h_j]$ | |

Table 8: The details of Probing inputs of *Probing task 3*. $d_i$ denotes the demand for node $i$.

| | | Probing input | # Observations | Features | Label |
|---|---|---|---|---|---|
| 20 and 100 | w/o ints. | AM-Init | | $[h_i, h_j]$ | |
| | | AM-Enc-*l*1 | | $[h_i, h_j]$ | |
| | | AM-Enc-*l*3 | | $[h_i, h_j]$ | |
| | | POMO-Enc-*l*1 | | $[h_i, h_j]$ | |
| | | POMO-Enc-*l*6 | 10000 | $[h_i, h_j]$ | $d_i + d_j$ |
| | | LEHD-Enc-*l*1 | | $[h_i, h_j]$ | |
| | | LEHD-Dec-*l*1 | | $[h_i, h_j]$ | |
| | | LEHD-Dec-*l*6 | | $[h_i, h_j]$ | |
| | w/ ints. | AM-Init | | $[h_i \odot h_j]$ | |
| | | AM-Enc-*l*1 | | $[h_i \odot h_j]$ | |
| | | AM-Enc-*l*3 | | $[h_i \odot h_j]$ | |
| | | POMO-Enc-*l*1 | | $[h_i \odot h_j]$ | |
| | | POMO-Enc-*l*6 | 10000 | $[h_i \odot h_j]$ | $d_i + d_j$ |
| | | LEHD-Enc-*l*1 | | $[h_i \odot h_j]$ | |
| | | LEHD-Dec-*l*1 | | $[h_i \odot h_j]$ | |
| | | LEHD-Dec-*l*6 | | $[h_i \odot h_j]$ | |

## B.3 Codes and Datasets for Reproducibility

We provide a GitHub repository[6] containing all codes required to construct the probing datasets. The repository includes: (1) instance generation with theoretical and greedy solutions, (2) scripts for extracting embeddings from different NCO models, (3) an example probing experiment with CS-Probing and visualization, and (4) training code for the LEHD regularization experiment in Section 4.5.

Although the repository already supports dataset generation, we plan to openly release all probing datasets to further facilitate research. This will save dataset preparation effort and enable more convenient usage.

---

[6]`https://github.com/123zhangzq/NeurIPS2025_probing/`

### B.4 Experiments Compute Resources

In this study, we use NVIDIA A100-40G GPU with AMD EPYC Milan 7713 CPU. It is important to note that the primary experimental processes involved in this study, namely training linear probing models or solving linear models using standard statistical methods (such as OLS for regression and MLE for classification), require minimal computational resources and can be completed within seconds.

However, collecting datasets does consume some computational resources. For instance, solving combinatorial optimization problems using commercial solvers typically requires CPU computation. Additionally, even when not training NCO models but merely performing inference to obtain NCO model embeddings, the process is also completed within seconds when using a GPU.

## C Detailed Results of All Probing Tasks

### C.1 Main Results of Four Probing Tasks

Tables 9 and 10 present the complete results of the four probing tasks. In addition to the discussions in Section 3.2, we provide more results from these tables and their corresponding discussions here.

#### C.1.1 *Probing Task 1* and *Probing Task 2*

**Knowledge existence.** For *Probing Task 1* (the Euclidean distance regression task), it is important to note that a linear model cannot directly capture the nonlinear relationship of Euclidean distance. Thus, a linear model would have no explanatory power if the input only consists of the nodes' coordinate information. In the most extreme case, where the input for *Probing Task 1* (i.e., the features) are solely the two nodes' coordinates, the regression model's $R^2$ value would be zero, because the covariance between the label and the linear model's output is zero. This result is also reflected in the experimental findings presented later in Section 3.2. On the other hand, when the probing model's $R^2$ value is greater than 0, and the closer it is to 1, the stronger the evidence that the NCO model has the ability to perceive Euclidean distances. This indicates that the information related to Euclidean distance, encoded in the model's embeddings, can be linearly extracted, thereby validating the NCO model's ability to effectively represent this information.

For *Probing Task 2*, to verify whether NCO models can avoid myopia rather than merely distinguish between different paths, we train a probing model using raw path features (i.e., the Euclidean distances of the paths) as input. This yields an AUC of 0.64, which is close to random guessing (0.5), in contrast to the AUC of 0.86 achieved when using NCO model embeddings as input. This result highlights that the model's behavior goes beyond simple path discrimination. Detailed results are shown in Figure 11.

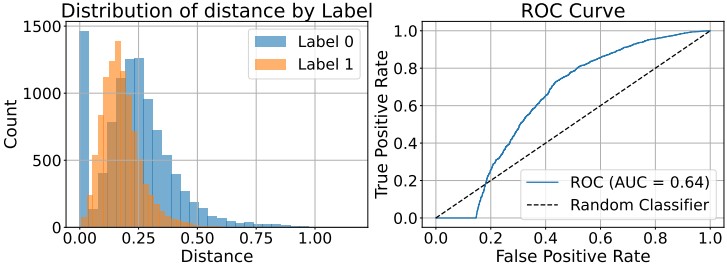

Figure 11: A classifier is trained using only the raw path feature (namely distance) as input. The left plot shows the label distribution, and the right plot presents the ROC results.

**More results about AM/POMO.** We observe differences in how different model architectures represent Euclidean distances. In the results shown in Table 9, for both 20-node and 100-node instances, LEHD's approach, utilizing a single encoder layer followed by multi-layer attention recalculations between the current node and other nodes in the decoder for each decision, demonstrates a more robust method for capturing information. From the table, we observe that LEHD achieves strong results regardless of the presence of interaction terms. In contrast, AM and POMO, which

Table 9: Comparison of probing task results for NCO models. The underlined results indicate they are derived from the final node embeddings of the three models.

| | | Probing input | Probing task 1 | | | Probing task 2 | | | | |
|---|---|---|---|---|---|---|---|---|---|---|
| | | | RMSE | MAE | $R^2$ score | Accuracy | Precision | Recall | F1 score | AUC |
| 20 node embeddings | w/o ints. | AM-Init | 0.2452 | 0.2028 | -0.0003 | 49.28% | 0.49 | 0.47 | 0.48 | 0.49 |
| | | AM-Enc-$l1$ | 0.2066 | 0.1665 | 0.2899 | 66.90% | 0.70 | 0.59 | 0.64 | 0.72 |
| | | AM-Enc-$l3$ | 0.2119 | 0.1711 | 0.2529 | 70.43% | 0.73 | 0.65 | 0.69 | 0.76 |
| | | AM-Enc-$l3$-w/c | 0.2140 | 0.1724 | 0.2381 | 69.30% | 0.72 | 0.63 | 0.67 | 0.75 |
| | | AM-Enc-$l3$-w/g | 0.2134 | 0.1721 | 0.2423 | 70.97% | 0.74 | 0.65 | 0.69 | 0.76 |
| | | POMO-Enc-$l1$ | 0.2115 | 0.1711 | 0.2558 | 64.50% | 0.68 | 0.56 | 0.61 | 0.67 |
| | | POMO-Enc-$l6$ | 0.2196 | 0.1787 | 0.1981 | 70.10% | 0.71 | 0.67 | 0.69 | 0.76 |
| | | POMO-SL-Enc-$l6$ | 0.2183 | 0.1770 | 0.2073 | 70.50% | 0.72 | 0.66 | 0.69 | 0.76 |
| | | POMO-Enc-$l6$-w/c | 0.2250 | 0.1809 | 0.1876 | 69.75% | 0.71 | 0.67 | 0.69 | 0.76 |
| | | POMO-Enc-$l6$-w/g | 0.2266 | 0.1827 | 0.1764 | 69.92% | 0.71 | 0.67 | 0.69 | 0.76 |
| | | LEHD-Enc-$l1$ | 0.2115 | 0.1719 | 0.2554 | 64.08% | 0.68 | 0.53 | 0.60 | 0.67 |
| | | LEHD-Dec-$l1$ | 0.0062 | 0.0046 | 0.9994 | 74.10% | 0.79 | 0.66 | 0.72 | 0.78 |
| | | LEHD-Dec-$l6$ | 0.0590 | 0.0451 | 0.9418 | 78.25% | 0.79 | 0.77 | 0.78 | 0.86 |
| | w/ ints. | AM-Init | 0.1318 | 0.1000 | 0.7111 | 51.95% | 0.52 | 0.47 | 0.50 | 0.52 |
| | | AM-Enc-$l1$ | 0.0235 | 0.0171 | 0.9908 | 70.28% | 0.74 | 0.63 | 0.68 | 0.77 |
| | | AM-Enc-$l3$ | 0.0657 | 0.0514 | 0.9282 | 75.90% | 0.78 | 0.73 | 0.75 | 0.83 |
| | | AM-Enc-$l3$-w/c | 0.0653 | 0.0512 | 0.9291 | 74.95% | 0.77 | 0.72 | 0.74 | 0.82 |
| | | AM-Enc-$l3$-w/g | 0.0660 | 0.0518 | 0.9275 | 75.67% | 0.78 | 0.72 | 0.75 | 0.83 |
| | | POMO-Enc-$l1$ | 0.0543 | 0.0430 | 0.9510 | 69.23% | 0.72 | 0.62 | 0.67 | 0.74 |
| | | POMO-Enc-$l6$ | 0.1119 | 0.0890 | 0.7917 | 78.88% | 0.79 | 0.80 | 0.79 | 0.86 |
| | | POMO-SL-Enc-$l6$ | 0.1044 | 0.0825 | 0.8186 | 76.35% | 0.78 | 0.74 | 0.76 | 0.84 |
| | | POMO-Enc-$l6$-w/c | 0.1189 | 0.0942 | 0.7732 | 78.97% | 0.79 | 0.80 | 0.79 | 0.86 |
| | | POMO-Enc-$l6$-w/g | 0.1192 | 0.0942 | 0.7722 | 78.90% | 0.79 | 0.79 | 0.79 | 0.87 |
| | | LEHD-Enc-$l1$ | 0.0424 | 0.0325 | 0.9701 | 66.88% | 0.71 | 0.57 | 0.63 | 0.72 |
| | | LEHD-Dec-$l1$ | 0.0069 | 0.0052 | 0.9992 | 74.12% | 0.79 | 0.67 | 0.72 | 0.79 |
| | | LEHD-Dec-$l6$ | 0.0592 | 0.0452 | 0.9415 | 78.55% | 0.80 | 0.77 | 0.78 | 0.86 |
| 100 node embeddings | w/o ints. | AM-Init | 0.2498 | 0.2084 | -0.0012 | 50.48% | 0.51 | 0.45 | 0.48 | 0.50 |
| | | AM-Enc-$l1$ | 0.2186 | 0.1791 | 0.2332 | 56.00% | 0.57 | 0.53 | 0.55 | 0.60 |
| | | AM-Enc-$l3$ | 0.2212 | 0.1800 | 0.2151 | 66.30% | 0.68 | 0.61 | 0.65 | 0.71 |
| | | AM-Enc-$l3$-w/c | 0.2245 | 0.1830 | 0.1915 | 67.10% | 0.69 | 0.62 | 0.65 | 0.72 |
| | | AM-Enc-$l3$-w/g | 0.2224 | 0.1806 | 0.2062 | 65.88% | 0.68 | 0.60 | 0.64 | 0.71 |
| | | POMO-Enc-$l1$ | 0.2210 | 0.1799 | 0.2166 | 57.60% | 0.59 | 0.53 | 0.55 | 0.62 |
| | | POMO-Enc-$l6$ | 0.2231 | 0.1825 | 0.2014 | 71.83% | 0.72 | 0.72 | 0.72 | 0.79 |
| | | POMO-SL-Enc-$l6$ | 0.2219 | 0.1809 | 0.2102 | 72.65% | 0.73 | 0.72 | 0.72 | 0.81 |
| | | POMO-Enc-$l6$-w/c | 0.2249 | 0.1818 | 0.1646 | 71.25% | 0.72 | 0.72 | 0.72 | 0.79 |
| | | POMO-Enc-$l6$-w/g | 0.2240 | 0.1817 | 0.1711 | 71.35% | 0.71 | 0.72 | 0.72 | 0.78 |
| | | LEHD-Enc-$l1$ | 0.2194 | 0.1796 | 0.2280 | 55.93% | 0.56 | 0.54 | 0.55 | 0.60 |
| | | LEHD-Dec-$l1$ | 0.0094 | 0.0068 | 0.9986 | 67.45% | 0.72 | 0.57 | 0.64 | 0.72 |
| | | LEHD-Dec-$l6$ | 0.0469 | 0.0370 | 0.9647 | 76.50% | 0.77 | 0.75 | 0.76 | 0.85 |
| | w/ ints. | AM-Init | 0.1334 | 0.1033 | 0.7143 | 51.82% | 0.52 | 0.47 | 0.49 | 0.53 |
| | | AM-Enc-$l1$ | 0.0262 | 0.0193 | 0.9890 | 63.80% | 0.66 | 0.57 | 0.61 | 0.68 |
| | | AM-Enc-$l3$ | 0.0444 | 0.0339 | 0.9684 | 69.08% | 0.72 | 0.63 | 0.67 | 0.76 |
| | | AM-Enc-$l3$-w/c | 0.0587 | 0.0463 | 0.9448 | 70.33% | 0.73 | 0.66 | 0.69 | 0.77 |
| | | AM-Enc-$l3$-w/g | 0.0447 | 0.0340 | 0.9679 | 69.15% | 0.72 | 0.63 | 0.67 | 0.75 |
| | | POMO-Enc-$l1$ | 0.0276 | 0.0212 | 0.9877 | 66.15% | 0.68 | 0.60 | 0.64 | 0.71 |
| | | POMO-Enc-$l6$ | 0.0802 | 0.0640 | 0.8968 | 72.47% | 0.72 | 0.73 | 0.73 | 0.80 |
| | | POMO-SL-Enc-$l6$ | 0.0797 | 0.0638 | 0.8980 | 73.60% | 0.74 | 0.73 | 0.74 | 0.81 |
| | | POMO-Enc-$l6$-w/c | 0.0807 | 0.0645 | 0.8923 | 72.28% | 0.72 | 0.73 | 0.72 | 0.80 |
| | | POMO-Enc-$l6$-w/g | 0.0806 | 0.0645 | 0.8927 | 72.42% | 0.72 | 0.73 | 0.73 | 0.80 |
| | | LEHD-Enc-$l1$ | 0.0421 | 0.0325 | 0.9716 | 61.82% | 0.63 | 0.59 | 0.61 | 0.66 |
| | | LEHD-Dec-$l1$ | 0.0075 | 0.0054 | 0.9991 | 67.20% | 0.72 | 0.57 | 0.63 | 0.73 |
| | | LEHD-Dec-$l6$ | 0.0468 | 0.0367 | 0.9648 | 77.00% | 0.78 | 0.76 | 0.77 | 0.85 |

Table 10: Results of *probing task 3* and *probing task 4*.

| | | Probing input | Probing task 3 | | | Probing task 4 | | | | |
|---|---|---|---|---|---|---|---|---|---|---|
| | | | RMSE | MAE | $R^2$ score | Accuracy | Precision | Recall | F1 score | AUC |
| 20 node embeddings | w/o ints. | AM-Init | 0.0000 | 0.0000 | 1.0000 | 49.35% | 0.49 | 0.52 | 0.51 | 0.50 |
| | | AM-Enc-$l1$ | 0.0088 | 0.0070 | 0.9945 | 74.12% | 0.74 | 0.74 | 0.74 | 0.81 |
| | | AM-Enc-$l3$ | 0.0273 | 0.0219 | 0.9471 | 77.03% | 0.76 | 0.79 | 0.77 | 0.84 |
| | | LEHD-Enc-$l1$ | 0.0038 | 0.0030 | 0.9990 | 71.78% | 0.72 | 0.72 | 0.72 | 0.79 |
| | | LEHD-Dec-$l1$ | 0.0100 | 0.0078 | 0.9929 | 69.73% | 0.71 | 0.66 | 0.69 | 0.75 |
| | | LEHD-Dec-$l6$ | 0.0366 | 0.0288 | 0.9047 | 75.92% | 0.75 | 0.78 | 0.76 | 0.84 |
| | w/ ints. | AM-Init | 0.0000 | 0.0000 | 1.0000 | 62.82% | 0.64 | 0.58 | 0.61 | 0.67 |
| | | AM-Enc-$l1$ | 0.0269 | 0.0209 | 0.9488 | 81.4% | 0.80 | 0.84 | 0.82 | 0.89 |
| | | AM-Enc-$l3$ | 0.0955 | 0.0767 | 0.3533 | 83.30% | 0.82 | 0.85 | 0.84 | 0.91 |
| | | LEHD-Enc-$l1$ | 0.0112 | 0.0087 | 0.9910 | 76.10% | 0.76 | 0.76 | 0.76 | 0.84 |
| | | LEHD-Dec-$l1$ | 0.0159 | 0.0125 | 0.9820 | 71.75% | 0.73 | 0.69 | 0.71 | 0.77 |
| | | LEHD-Dec-$l6$ | 0.0632 | 0.0502 | 0.7169 | 77.78% | 0.76 | 0.80 | 0.78 | 0.87 |
| 100 node embeddings | w/o ints. | AM-Init | 0.0000 | 0.0000 | 1.0000 | 49.83% | 0.50 | 0.54 | 0.52 | 0.50 |
| | | AM-Enc-$l1$ | 0.0137 | 0.0110 | 0.9878 | 77.95% | 0.78 | 0.78 | 0.78 | 0.86 |
| | | AM-Enc-$l3$ | 0.0237 | 0.0187 | 0.9635 | 78.08% | 0.77 | 0.80 | 0.78 | 0.86 |
| | | POMO-Enc-$l1$ | 0.0199 | 0.0157 | 0.9743 | 72.72% | 0.74 | 0.70 | 0.72 | 0.81 |
| | | POMO-Enc-$l6$ | 0.0445 | 0.0356 | 0.8710 | 70.12% | 0.70 | 0.70 | 0.70 | 0.77 |
| | | LEHD-Enc-$l1$ | 0.0052 | 0.0040 | 0.9950 | 65.75% | 0.67 | 0.63 | 0.65 | 0.71 |
| | | LEHD-Dec-$l1$ | 0.0069 | 0.0055 | 0.9913 | 66.22% | 0.67 | 0.63 | 0.665 | 0.71 |
| | | LEHD-Dec-$l6$ | 0.0178 | 0.0140 | 0.9426 | 77.18% | 0.76 | 0.79 | 0.77 | 0.85 |
| | w/ ints. | AM-Init | 0.0000 | 0.0000 | 1.0000 | 55.80% | 0.56 | 0.51 | 0.54 | 0.58 |
| | | AM-Enc-$l1$ | 0.0356 | 0.0278 | 0.9177 | 85.79% | 0.84 | 0.89 | 0.86 | 0.92 |
| | | AM-Enc-$l3$ | 0.0645 | 0.0510 | 0.7290 | 87.50% | 0.85 | 0.91 | 0.88 | 0.93 |
| | | POMO-Enc-$l1$ | 0.0272 | 0.0214 | 0.9517 | 78.72% | 0.78 | 0.80 | 0.79 | 0.87 |
| | | POMO-Enc-$l6$ | 0.1157 | 0.0951 | 0.1281 | 72.67% | 0.72 | 0.74 | 0.73 | 0.80 |
| | | LEHD-Enc-$l1$ | 0.0069 | 0.0053 | 0.9915 | 66.57% | 0.66 | 0.68 | 0.67 | 0.72 |
| | | LEHD-Dec-$l1$ | 0.0086 | 0.0068 | 0.9867 | 66.15% | 0.67 | 0.64 | 0.65 | 0.71 |
| | | LEHD-Dec-$l6$ | 0.0308 | 0.0243 | 0.8280 | 78.60% | 0.77 | 0.81 | 0.79 | 0.86 |

embed all nodes through multiple encoder layers once, rely more heavily on the interaction terms between the embeddings of the two nodes when perceiving Euclidean distances.

As shown in the results for "AM-Enc-$l3$-w/c" and "AM-Enc-$l3$-w/g," even with the extra information provided by context embeddings or glimpse embeddings, AM and POMO do not improve the accuracy of perceiving the Euclidean distance between the current node and other nodes. Rows of "POMO-SL-Enc-$l6$" represents the embeddings of a POMO model trained using supervised learning (SL). The results show that the SL-trained POMO achieves similar probing task results to the RL-trained POMO. This observation aligns with the findings from the ablation study in [13], where SL-trained and RL-trained POMO models exhibit comparable performance.

**Results by model layer.** As shown in Figure 2(a), the initial embeddings (obtained by linearly projecting the coordinates into a high-dimensional space) of all three models exhibit weak Euclidean distance perception. However, after passing through just one attention layer, all models achieve highly accurate Euclidean distance perception. This ability slightly diminishes as model depth increases.

Despite this slight decline in Euclidean distance perception, NCO models learn additional capabilities that enable better decision-making. For instance, the ability to avoid myopic node selection improves with increased model depth, as illustrated in Figure 2(b). An exception is observed in the last two layers of LEHD, where the ability to avoid myopic decisions slightly decreases, potentially indicating that the model has learned more complex decision-making strategies. Future research could further explore this phenomenon and what LEHD learns in its deeper layers. Overall, through these two probing tasks, we demonstrate that when NCO models solve TSP problems, they can perceive Euclidean distances (low-level features) in shallow layers and learn a decision space beyond the Euclidean distance space (high-level features) in deeper layers. In this decision space, NCO models can avoid making myopic decisions.

By comparing the results (including both node-scale instances and whether interaction terms are used) of the same model across different layers, we find that the ability of the embeddings to perceive Euclidean distances decreases as the number of attention layers increases in all three models. Notably, after six attention layers, POMO shows a more significant decline in Euclidean distance perception compared to AM, which has the same structure but only three attention layers. This suggests that

while deeper attention layers may enhance other decision-making capabilities (as discussed in the *Probing Task 2*), the model's ability to perceive distances diminishes.

In subsequent research based on AM/POMO models, some models introduce node distance information to enhance performance: either by explicitly incorporating distance information to adjust the model's output [45], or by designing distance-aware attention mechanisms [46]. Through probing experiments, we verify that these approaches introduce Euclidean distance to mitigate its perception deficiency as the number of layers increases in NCO models. This provides important guidance for future improvements to AM and POMO-based models.

### C.1.2 *Probing Task 3* and *Probing Task 4*

This section provides a detailed discussion of Section 3.2, specifically addressing the question: ***Do NCO embeddings encode constraint information, as demonstrated with the CVRP?***

*Probing Task 3*: **Capacity constraint.** Through *Probing Task 3*, using the capacity constraint in the CVRP problem as an example, we demonstrate that probing can be applied to study the ability of NCO models to represent low-level information related to constraints. From Table 10, we can see that the embeddings of NCO models unquestionably contain the linear information of demand. This is particularly evident in the initial embeddings, where the three-dimensional raw features (i.e., x and y coordinates and demand) are directly projected into a high-dimensional space, allowing the demand information to be fully extracted ($R^2 = 1$). Even after the embeddings undergo attention mechanisms and the high-dimensional features of nodes are fused, this information remains largely extractable, with $R^2$ values ranging from above 0.7 to nearly 1.

We observe that, while all three NCO models can capture the linear (additive) relationship between node demands, this ability weakens with an increasing number of layers, similar to the perception of Euclidean distances. This observation is particularly noteworthy in the Hadamard product probing input, $[h_i \odot h_j]$. As discussed in Section B.1.3, we simulate attention calculations using this Hadamard product input. Many NCO models, including AM and POMO, calculate a compatibility score by attention calculations before applying the output Softmax. In this context, the $R^2$ values for the final output layer decrease significantly compared to the first layer, as shown in the "w/ ints." rows in Table 10. In some results, $R^2$ even drops to the 0.1 to 0.35 range, indicating that these NCO models may no longer accurately capture whether the demand exceeds vehicle capacity and, as a result, are unable to actively select a feasible next node. Instead, they rely passively on masking to enforce final output modifications and constraints.

*Probing Task 4*: **Same route.** Here, we take the example of using probing to explore whether the embeddings of the AM model can encode information about whether two nodes belong to the same route in the optimal solution, providing a detailed analysis of how this conclusion is reached. The same reasoning process applies to other NCO models, with detailed results available in Table 10.

Figure 12 presents the AUC results for *Probing Task 4* using different input data for the probe. The results in Figures (a) and (b) (with AUC values close to 0.5) rule out the possibility that the two nodes can be linearly separated solely based on their Euclidean distance or their node coordinates. Therefore, if the embeddings learned by NCO models can be linearly separated by the probe (i.e., AUC larger than 0.5, approaching 1), it indicates that the embeddings contain information about whether two nodes should belong to the same route. As shown in Figures (c) and (d), when using AM embeddings as the probing input, the AUC is significantly greater than 0.5, demonstrating that AM can effectively encode information for determining whether two nodes should belong to the same route when solving the CVRP. Other NCO models also exhibit this capability, with detailed results provided in Table 10.

**Discussion.** In the capacity constraint probing tasks, we explored the decision rationales of two NCO models in handling this constraint. The probing results show that NCO models can perceive linearly additive demand information as well as more abstract decision-supporting information. While these experiments reveal how NCO models handle capacity constraints, unlike the previous probing tasks on TSP, these results do not show a strong correlation with the models' final performance. For instance, in *Probing Task 4*, AM achieved the best probing results, yet it is not the best-performing model on the CVRP.

These two probing tasks raise interesting research questions regarding the design of NCO models for handling constraints in the future. Should additional constraint-related information be incorporated

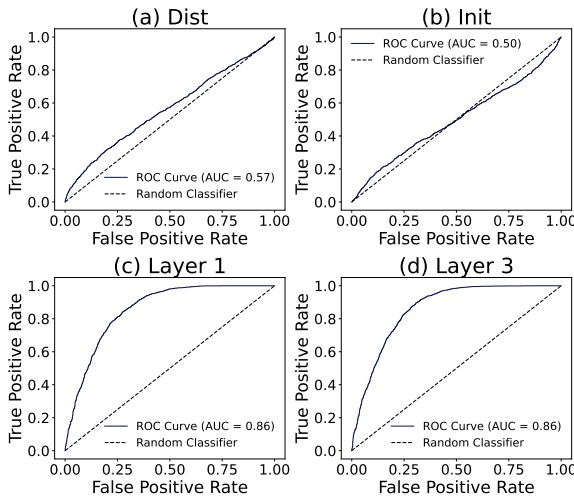

Figure 12: AUC results for *Probing Task 4*. The inputs from (a) to (d) are: (a) the Euclidean distance between two nodes, (b) the initial embedding obtained by directly projecting the raw features into a high-dimensional space (AM-Init), (c) the embedding after the first attention layer (AM-Enc-$l1$), and (d) the embedding after the final attention layer (AM-Enc-$l3$)

into NCO models, similar to how distance information is integrated in previous studies [45, 46]? Are there other decision-related features in CVRP that exhibit a stronger correlation with model performance and can help better understand the decision-making process of NCO models? Future work can further explore these questions by designing and implementing additional probing tasks to deepen our understanding of how NCO models handle constraints.

## C.2 Robustness Check

### C.2.1 Other Models

We conduct a preliminary experiment on DIFUSCO [18], a diffusion-based NCO model. After training the model for 20 epochs, we apply probing to analyze its learned representations. Table 11 presents the node embedding results. As the number of GNN layers increases, the capacity to capture Euclidean distances slightly decreases, whereas the ability to identify optimal edges improves. This distinction between low-level and high-level feature encoding is consistent with the patterns observed in transformer-based models discussed above. These results demonstrate that probing is also effective for analyzing alternative architectures, such as diffusion-based models where the GNN serves as the denoising network.

Table 11: Probing results on node embeddings of DIFUSCO. Here, h_init denotes the initial embeddings, and h_12 denotes the embeddings after 12 GNN layers.

| Probing input | *Probing Task 1* | *Probing Task 2* |
|---|---|---|
| h_init | 0.9476 | 0.49 |
| h_12 | 0.8710 | 0.73 |

### C.2.2 Distance Perception in Non-Euclidean Space

To further validate the robustness of probing as a tool for analyzing NCO models and the probing tasks we designed, we demonstrate *Probing Task 1* for distance perception in non-Euclidean space. Specifically, we selected MatNet [47], a state-of-the-art model designed for solving asymmetric TSP.

We use MatNet's row embeddings and column embeddings for pairs of nodes as features, and the distances between the corresponding nodes in the distance matrix as labels to construct the probing dataset. For example, the row embeddings of node $i$ and node $j$ are used as features, with the

corresponding label being the value in the distance matrix at the intersection of row $i$ and column $j$, denoted as $dist(i, j)$. Similarly, the column embeddings of node $i$ and node $j$ are used as features, with the label being $dist(j, i)$.

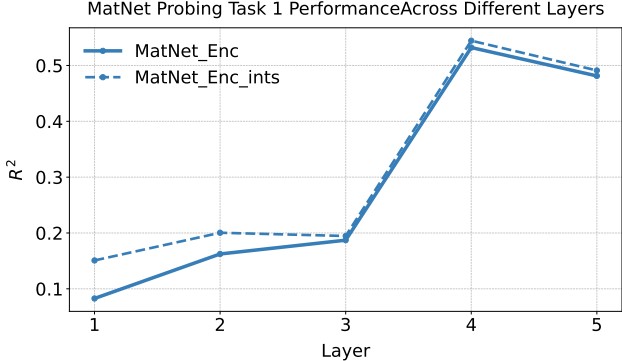

Figure 13: Probing results of MatNet across different layers.

The results are shown in Figure 13. As the number of layers in MatNet increases, the ability of its embeddings to perceive distances improves, with the $R^2$ rising from less than 0.2 in the first layer to approximately 0.5 in the final layer. Additionally, we conduct supplementary comparison experiments. In the first experiment, serving as a baseline, the embeddings of node $i$ and node $j$ are used as inputs, but the labels are replaced with random distance values unrelated to both nodes from the distance matrix. The resulting probing $R^2$ is -0.0232, indicating that the probe could not learn any distance information from random labels based on the embeddings. In the second experiment, we swap the labels between row and column embeddings, assigning the row embeddings of node $i$ and $j$ with the label $dist(j, i)$ and vice versa. The resulting probing $R^2$ is 0.2532. Comparing these results, we conclude the following: MatNet's dual-attention structure effectively learns information from the asymmetric distance matrix. Furthermore, regardless of whether the embeddings of two nodes are correctly aligned, they can still partially represent distance information. However, the model's ability to capture correct distance information between two nodes is significantly stronger than its ability to capture incorrect distance information, with $R^2$ values of approximately 0.5 versus 0.2, respectively.

### C.2.3 JSSP Precedence Constraint

In addition to the routing problem analyzed earlier, we also apply probing to test the precedence constraints in the Job-shop Scheduling Problem. For JSSP, we evaluate a classic learning model [48], which is based on a graph neural network. The datasets for this probing task are constructed as follows: we extract embeddings for all nodes, pair two node embeddings that satisfy the precedence constraint with a label of 1 ($[h_i, h_j]$-1), and pair two node embeddings that violate the constraint with a label of 0 ($[h_m, h_n]$-0). As an ablation, we also construct an alternative dataset where pairs that satisfy the precedence constraint are incorrectly labeled as 0: $[h_i, h_j]$-1, $[h_n, h_m]$-0.

The results show that for the correct dataset, the probing model achieves an AUC of 1, while for the ablation dataset, the AUC is 0.5. This indicates that the NCO model effectively captures the precedence constraint information between nodes in its embeddings. Here, we provide an initial demonstration of how probing can explore the NCO model's perception of constraints in the JSSP. In the future, more sophisticated probing tasks can be designed to further analyze how the NCO model perceives constraints and incorporates them into its decision-making process, thereby offering deeper insights into the design of NCO models.

### C.3 Probing NCO Model Performance

The two probing tasks discussed in the previous section not only validate that NCO models can embed decision-related information at different levels but also provide preliminary evidence that probing can be used to explore the performance of NCO models. For example, in *Probing Task 2* with 20-node instances, the probing results show that AM (with an AUC of 0.83) slightly underperforms compared to POMO and LEHD (both 0.86). In 100-node instances, the probing results rank from

lowest to highest as AM (0.76), POMO (0.80), and LEHD (0.85). These probing results for both 20-node and 100-node instances are consistent with the final performance of these models in solving TSP problems of the same sizes. Specifically, for 20 nodes, POMO and LEHD perform similarly and slightly outperform AM, whereas for 100 nodes, LEHD outperforms POMO, which in turn outperforms AM (see the greedy inference methods results in the Table 2 in [12] and Table 1 in [13]).

This section further explores how probing can be used to study the impact of NCO models' representational capabilities on their performance from multiple perspectives. Additionally, we introduce CS-Probing to examine the differences between internal representations and inductive biases across various NCO model architectures, providing direct evidence to explain generalization performance. For more details, please refer to Section 4. Here, we only provide the results of probing tasks as an indirect perspective to explore the generalization of NCO models.

**Generalization to larger scale.** One of the advantages of the LEHD model is its superior generalization performance on large-scale problems compared to AM and POMO. To validate this, we create a dataset with 200 node instances. Based on this dataset, we obtained embeddings from three NCO models pretrained on 100-node problems when solving 200-node problems.

Table 12: Experimental results for the 200 node instances.

| | Probing input | Probing task 1 ($R^2$) | Probing task 2 (AUC) |
|---|---|---|---|
| w/o ints. | AM-Enc-$l3$ | 0.1673 | 0.71 |
| | POMO-Enc-$l6$ | 0.1352 | 0.80 |
| | LEHD-Dec-$l6$ | 0.9563 | 0.86 |
| w/ ints. | AM-Enc-$l3$ | 0.9458 | 0.76 |
| | POMO-Enc-$l6$ | 0.9100 | 0.80 |
| | LEHD-Dec-$l6$ | 0.9588 | 0.86 |

From Table 12, we observe that the three NCO models pretrained on 100-node instances exhibit varying performance on the two probing tasks for 200-node instances. Notably, for *Probing Task 2*, which explores information more directly related to final decisions (distinguishing optimal edges from myopic ones), the results are fully consistent with their performance on the optimization problem outcomes, i.e., LEHD outperforms POMO, which in turn outperforms AM. Next, we conduct additional experiments to explore how the unique structure of LEHD enhances its representational capability, enabling it to better focus on nodes to be selected.

**Further experiments on LEHD.** LEHD's recalculation of the embeddings of candidate nodes in its decoder, through the attention mechanism with the current node embedding, may allow it to more effectively capture the relationships between the current node and other nodes. Specifically, as shown in the decoder of Figure 1(b), the embedding of the current node, $h_s$, participates in the attention calculations with the remaining nodes after passing through a linear projection, updating their embeddings. In contrast to AM and POMO, which treat all node embeddings equally and perform node embedding only once, LEHD's decoder design allows for a more accurate perception of the distances between the current node and the remaining nodes. To verify this, we conducted additional experiments on LEHD, and the results are presented in Table 13.

Table 13: Experiments for LEHD. The first two rows show distance perception between non-current nodes and others, while the third row shows the effect of removing attention from LEHD.

| Probing input | RMSE | MAE | $R^2$ score |
|---|---|---|---|
| LEHD-Dec-$l1$-other | 0.2091 | 0.1694 | 0.2620 |
| LEHD-Dec-$l6$-other | 0.2318 | 0.1898 | 0.0927 |
| LEHD-Dec-w/o-att | 0.2115 | 0.1719 | 0.2555 |
| LEHD-Dec-$l6$ | 0.0590 | 0.0451 | 0.9421 |

First, we extract the embeddings of two remaining nodes (i.e., nodes to be selected) for probing and find that the probe achieves an $R^2$ of only 0.0927. This indicates that LEHD is indeed more focused on the relationship between the current node and other nodes. Additionally, when we probe the embedding from the linear projection below $h_s$ in the decoder (Figure 1(b), before the attention calculation), its $R^2$ dropped to 0.2555, significantly lower than the original 0.9421. This suggests

that the attention mechanism in LEHD's decoder is crucial for accurately capturing the Euclidean distances between the current node and the remaining nodes (i.e., nodes to be selected).

This leads to an insight for future NCO model: recalculating node embeddings through the attention mechanism in the decoder enables more accurate perception of Euclidean distances than relying solely on context embeddings, as in the case of AM and POMO, to provide current information (more details in Appendix C.1.1). To further validate this, we next examine probing from an entirely new perspective.

# D    CS-Probing for NCO

## D.1    LEHD Training

Figure 14 shows the CS-Probing results of the LEHD model at different training epochs. From the results, we observe that in the early stages of training, LEHD does not predominantly capture decision-related knowledge within a few specific dimensions. However, as training progresses, the model increasingly tends to fixate on a small number of dimensions to encode this knowledge. This tendency is evident from the larger absolute values of probing model coefficients associated with certain dimensions, as well as the increased disparity between the coefficients of different dimensions. Eventually, after sufficient training, LEHD exhibits the inductive bias shown in Figure 5 (e) and (f).

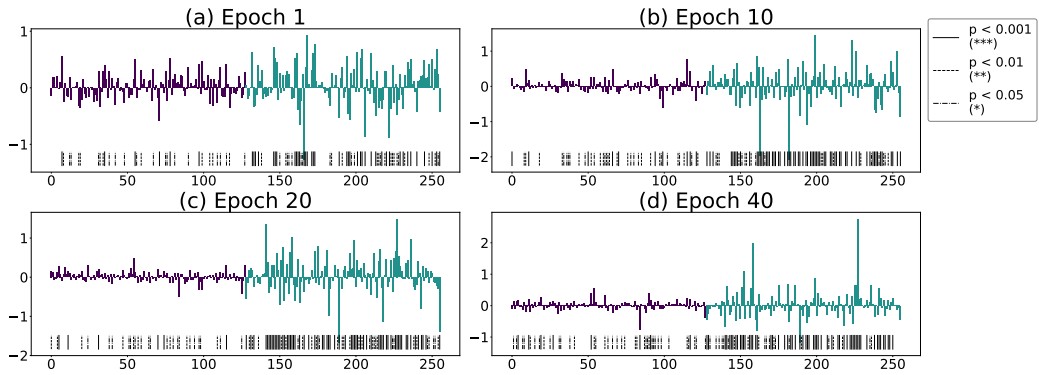

Figure 14: CS-Probing results of LEHD at different training epochs.

## D.2    Visualizing CS-Probing Outcomes for NCO Generalization

Figure 15 presents the CS-Probing analysis for the three NCO models, highlighting the key embedding dimensions (top-5) identified during generalization. These three figures provide a visualization of the data presented in Table 2. In each figure, the four sub-figures present the probing results of the specific NCO model on *Probing Task 1* (left two plots) and *Probing Task 2* (right two plots), showing both the probe coefficients and their statistical significance. For each task, the top plot shows the result of the NCO model trained and tested on 20-node instances, while the bottom plot shows the result of the same model (training on 20-node instances) generalized to 100-node test instances. In each plot, the left section (in purple) represents the results for 128-dimensional embedding of the current node, and the right section (in blue) represents the results for 128-dimensional embeddings of the candidate nodes—together forming a 256-dimensional results. Below each dimension is its significance level. Red vertical lines highlight the top-n dimensions by absolute coefficient magnitude (n = 1 to 5), with their ranks labeled above the lines. Note: if two top-n dimensions are close, the rank labels are slanted to avoid overlap.

## D.3    Generalization of AM and POMO on Near Out-of-Distribution Data

In Section 4.3, we investigate the internal mechanisms underlying the generalization ability of the three models by analyzing whether they consistently retain specific knowledge dimensions during generalization. The results indicate that models with superior generalization performance tend to consistently use a fixed set of dimensions to capture certain knowledge, reflecting the transferability of

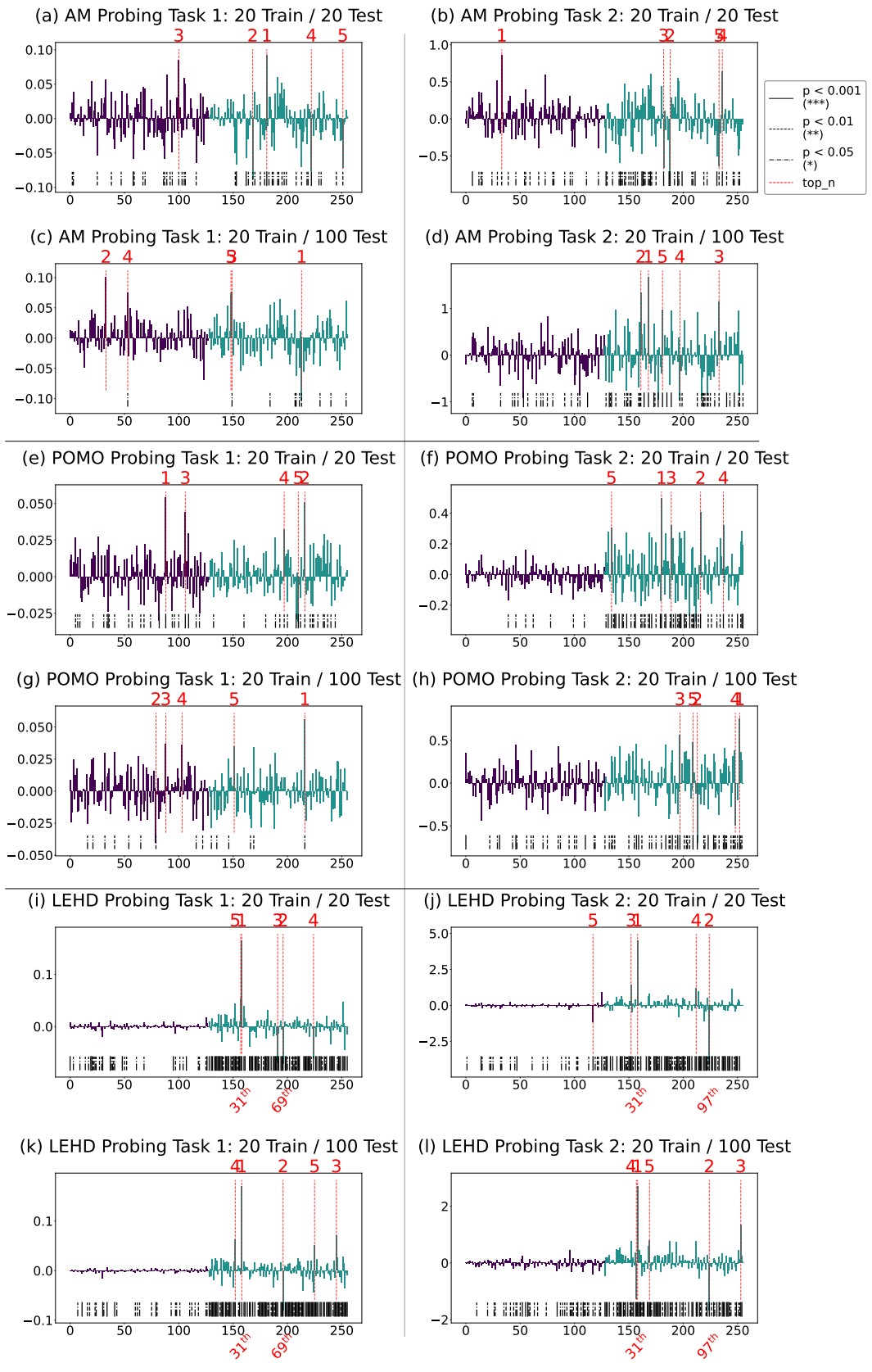

Figure 15: CS-Probing results for the two probing tasks across three models, including both in-distribution and out-of-distribution generalization results. This corresponds to the visualization of the data presented in Table 2.

learned representations. Conversely, when the knowledge encoded in specific embedding dimensions becomes disorganized during generalization, the model's performance deteriorates.

Table 14: Top 5 dimensions from CS-Probing results two TSP probing tasks on TSP-20 and TSP-21 across AM and POMO models.

| | | Top_n | *20 train / 20 test* | | | *20 train / 21 test* | | |
|---|---|---|---|---|---|---|---|---|
| | | | Dim. | Coef. | Sig. level | Dim. | Coef. | Sig. level |
| Probing Task 1 | AM | 1 | **54 (candidate node)** | 0.0917 | *** | **101 (current node)** | 0.0803 | * |
| | | 2 | 41 (candidate node) | -0.0884 | ** | 90 (current node) | 0.0752 | ** |
| | | 3 | **101 (current node)** | -0.0848 | * | **54 (candidate node)** | 0.0729 | ** |
| | | 4 | 95 (candidate node) | -0.0767 | *** | 48 (candidate node) | -0.0631 | *** |
| | | 5 | 124 (candidate node) | -0.0712 | ** | 35 (candidate node) | -0.0617 | *** |
| | POMO | 1 | **89 (current node)** | 0.0539 | *** | **89 (current node)** | 0.0615 | *** |
| | | 2 | **89 (candidate node)** | 0.0510 | * | **89 (candidate node)** | 0.0516 | ** |
| | | 3 | **107 (current node)** | 0.0443 | *** | **107 (current node)** | 0.0389 | ** |
| | | 4 | 70 (candidate node) | 0.0322 | ** | 15 (candidate node) | -0.0376 | * |
| | | 5 | 83 (candidate node) | -0.0308 | *** | 74 (current node) | 0.0352 | *** |
| Probing Task 2 | AM | 1 | 34 (current node) | 0.8561 | ** | **61 (candidate node)** | -0.9994 | *** |
| | | 2 | **61 (candidate node)** | -0.8153 | *** | 125 (candidate node) | -0.7887 | *** |
| | | 3 | 55 (candidate node) | -0.6612 | *** | 109 (candidate node) | 0.7307 | *** |
| | | 4 | 109 (candidate node) | 0.6406 | *** | 55 (candidate node) | -0.7019 | *** |
| | | 5 | 106 (candidate node) | -0.6329 | ** | 124 (candidate node) | -0.6858 | *** |
| | POMO | 1 | **53 (candidate node)** | 0.5004 | *** | **53 (candidate node)** | 0.4796 | *** |
| | | 2 | **89 (candidate node)** | 0.4088 | *** | 74 (candidate node) | 0.3944 | *** |
| | | 3 | 62 (candidate node) | 0.3238 | *** | 27 (candidate node) | 0.3804 | ** |
| | | 4 | 110 (candidate node) | 0.3217 | *** | **89 (candidate node)** | 0.3182 | ** |
| | | 5 | 7 (candidate node) | 0.3081 | *** | 42 (candidate node) | 0.3067 | *** |

To further validate this observed phenomenon and conduct a more in-depth exploration, we evaluate the CS-Probing results of AM and POMO when generalizing to datasets with smaller distributional differences—specifically, those on which they exhibit better generalization performance. We train the models on 20-TSP and test them on both 20-TSP and a similar distribution, 21-TSP. Table 14 presents the CS-Probing results for AM and POMO on both datasets. The results indicate that both models consistently use the same dimensions to capture knowledge across the two tasks.

Comparing these findings with Table 2, we observe that models capable of generalization—such as LEHD when scaling up, or AM and POMO when generalizing to slightly different distributions—consistently demonstrate the aforementioned characteristic. This further reinforces our claim that the ability to consistently use key dimensions during generalization is indicative of robust performance.

## D.4 Other Random Seeds

For the instance of random seed 2 (upper subplot of Figure 16), the current node is node 4. The myopic choice based on Euclidean distance would be node 2, while the correct choice in the optimal solution is node 13. In the two identified key dimensions, node 4 is indeed closer to node 13 than to node 2.

For the instance of random seed 3 (lower subplot of Figure 16), the current node is node 8. The myopic choice based on Euclidean distance would be node 1, while the correct choice in the optimal solution is node 9. In the two identified key dimensions, node 8 is indeed closer to node 9 than to node 1, even though the difference is small. This is consistent with the node distribution shown in the left two plots, where nodes 9 and 1 are very close to each other, and their distances to node 8 differ only slightly.

## D.5 CS-Probing for Probing Task 3 and 4

The aforementioned CS-Probing analysis for the two TSP-related probing tasks can similarly be applied to the two CVRP-related probing tasks. Here, we take the evaluation of generalization ability as an example, using models trained on 20-node instances and testing them on both 20-node and 100-node datasets. Since POMO does not provide a model trained on 20-node instances, we only test the other two NCO models. The experimental results are presented in Table 15.

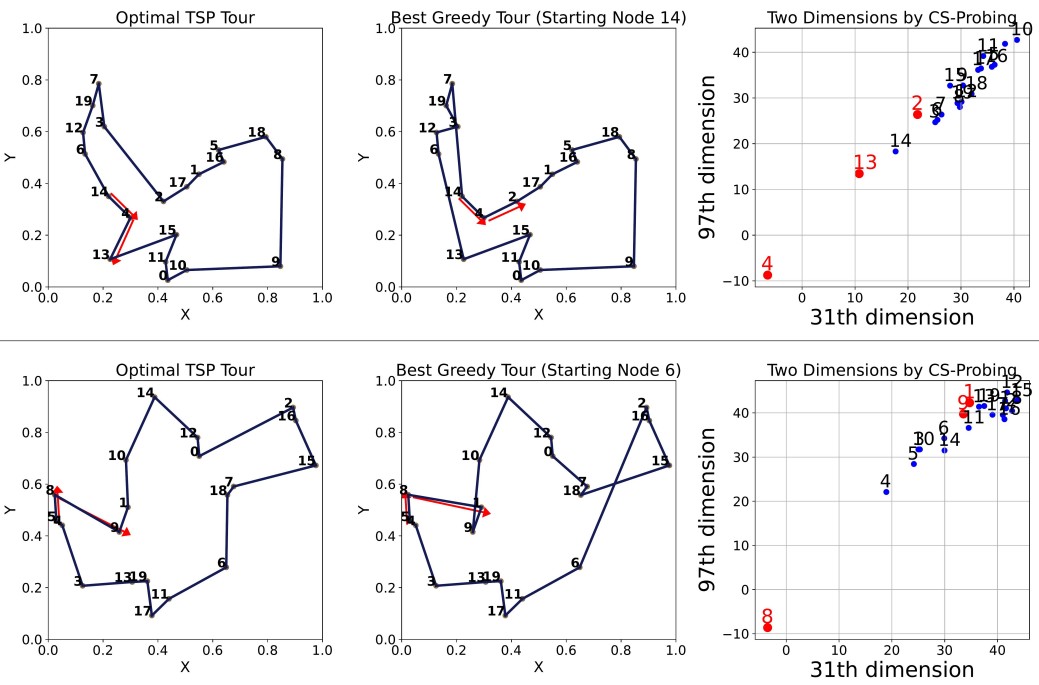

Figure 16: Solution results and key 2-d dimensions in node embedding.

Table 15: Top 5 dimensions from CS-Probing results two CVRP probing tasks across AM and LEHD models. (POMO did not provide the pre-trained model on 20-CVRP)

| | | Top_n | *20 train / 20 test* | | | *20 train / 100 test* | | |
|---|---|---|---|---|---|---|---|---|
| | | | Dim. | Coef. | Sig. level | Dim. | Coef. | Sig. level |
| Probing Task 3 | AM | 1 | **61 (candidate node)** | 0.0729 | *** | **61 (current node)** | 0.1123 | *** |
| | | 2 | **61 (current node)** | 0.0691 | *** | **61 (candidate node)** | 0.0998 | *** |
| | | 3 | **45 (candidate node)** | -0.0631 | *** | 106 (current node) | -0.0661 | *** |
| | | 4 | 45 (current node) | -0.0624 | *** | **45 (candidate node)** | -0.0647 | *** |
| | | 5 | **67 (current node)** | -0.0464 | *** | **67 (current node)** | 0.0639 | *** |
| | LEHD | 1 | **30 (candidate node)** | -0.0325 | *** | **30 (current node)** | -0.0262 | *** |
| | | 2 | **30 (current node)** | -0.0322 | *** | **30 (candidate node)** | -0.0223 | *** |
| | | 3 | 118 (candidate node) | -0.0221 | *** | 113 (current node) | 0.0195 | *** |
| | | 4 | 118 (current node) | -0.0202 | *** | 113 (candidate node) | 0.0182 | *** |
| | | 5 | 5 (current node) | 0.0202 | *** | 116 (current node) | -0.0161 | *** |
| Probing Task 4 | AM | 1 | **45 (candidate node)** | -1.5845 | *** | 87 (candidate node) | -2.2114 | *** |
| | | 2 | 38 (candidate node) | 1.1667 | *** | 110 (current node) | -1.9473 | *** |
| | | 3 | 80 (candidate node) | -1.1331 | *** | 87 (current node) | 1.9408 | *** |
| | | 4 | 59 (candidate node) | -1.0980 | *** | 40 (candidate node) | 1.8984 | *** |
| | | 5 | 21 (candidate node) | -1.0775 | *** | **45 (candidate node)** | 1.8825 | *** |
| | LEHD | 1 | 1 (candidate node) | -0.7821 | *** | 71 (candidate node) | 1.4377 | *** |
| | | 2 | 8 (candidate node) | 0.6919 | *** | 30 (candidate node) | 1.1433 | *** |
| | | 3 | 75 (candidate node) | 0.6908 | *** | 2 (candidate node) | -1.0100 | *** |
| | | 4 | 76 (candidate node) | 0.6904 | *** | 99 (candidate node) | 1.0020 | *** |
| | | 5 | 102 (candidate node) | 0.6737 | *** | 30 (current node) | -0.9591 | *** |

The results indicate that during generalization, NCO models can effectively capture linearly additive demand information, demonstrating a strong ability to represent linearly additive knowledge. However, when it comes to more complex, abstract high-level information related to the global optimal solution, the learned dimensions become disorganized. This suggests that while NCO models can robustly encode simple additive information, their ability to maintain structured representations of more complex knowledge requires further investigation.

In the future, designing more CVRP-related probing tasks and conducting CS-Probing experiments can help gain a deeper understanding of the internal mechanisms of NCO models on CVRP, such as how they perceive complex constraints.

## E   Limitations

One potential limitation of using probing to explore NCO models is the cost associated with collecting probing datasets. The first challenge is the runtime cost. Regarding the data collection process in our study, we summarize the time required to collect 10,000 instances in Table 16. As shown, although NCO model training typically takes several days to a week, we find this data collection cost to be reasonable.

Table 16: Computational cost of NCO training and optimization solver at different scales.

|  | 20-TSP | 100-TSP |
| --- | --- | --- |
| POMO Training | 1–2 days | 2–3 days |
| Exact Solver | 1–2 hrs | 2–3 days |

Another limitation arises when probing requires the optimal solutions of instances to verify whether the model captures specific knowledge, as commercial solvers are not freely available. Moreover, even when commercial solvers are accessible, this requirement can still be challenging for large-scale and complex combinatorial optimization problems. For instance, solving the CVRP with 100 nodes using exact solvers like Gurobi is computationally infeasible. To address this challenge, we employ validated heuristic-based solution methods, such as the HGS algorithm. These open-source methods provide practical alternatives for researchers who may lack access to commercial solvers.

Lastly, we plan to release all datasets used in this study to facilitate future research. We believe that as more studies propose new probing tasks and make their datasets publicly available, it will contribute to advancing future research within the NCO community, promoting the transparency and interpretability of black-box NCO models.

