# OpenReview forum: "Probing Neural Combinatorial Optimization Models"
_NeurIPS.cc/2025/Conference — NeurIPS 2025 spotlight_

### Official Review · Reviewer_veN7 · 2025-06-30

**Clarity:** 3
**Significance:** 2
**Originality:** 2
**Rating:** 4
**Confidence:** 3

**Summary:**

This paper uses probing techniques to analyze the inner workings of neural combinatorial optimization (NCO) methods: the authors suggest that while these methods have achieved remarkable success, we lack an understanding of the reasons for this success particularly in terms of model representation and reasoning rationale. Therefore, the paper introduces Coefficient Significance Probing (CS-probing) to investigate the embedding space of transformer-based NCO models. The key finding is that models with better generalization tend to consistently rely on the same embedding dimensions across tasks. In contrast, models whose knowledge becomes disorganized across dimensions during generalization tend to suffer performance degradation.

**Questions:**

Recent works have introduced diffusion models for combinatorial optimization [1,2]. Could the proposed CS-probing technique be applied to analyze such models as well?

[1] DIFUSCO: Graph-based Diffusion Solvers for Combinatorial Optimization, NeurIPS 2023


[2] A Diffusion Model Framework for Unsupervised Neural Combinatorial Optimization, ICML 2024

**Ethical Concerns:**

["NO or VERY MINOR ethics concerns only"]

**Final Justification:**

The comprehensive rebuttal and new experimental results have addressed most of my concerns. I increased my score accordingly

**Limitations:**

yes

**Quality:**

2

**Strengths And Weaknesses:**

Strengths And Weaknesses:

Strengths:

1. This is the first work to apply probing techniques in the context of NCO, which helps reveal the internal mechanisms of these models.

2. The paper is clearly written and well-structured.

Weaknesses:

1. The paper is purely empirical with no methodological innovation. While this is acceptable, the experimental setup is limited in scope—focused only on small-scale instances of TSP, CVRP, and JSSP. Broader empirical evaluation is necessary to draw robust conclusions.

2. It is difficult to pinpoint where differences between model behaviors originate. Are they due to variations in neural architectures or differences in training algorithms?

3. The conclusions drawn from CS-probing, while interesting, lack practical implications. It remains unclear how these insights could inform the design of improved NCO methods in future work.

4. The benchmarked methods, such as POMO and AM, are relatively early approaches. It would be more compelling to evaluate with newer, state-of-the-art models.

Minor comment: The paper relies heavily on the appendix. I understand that the page limit makes it difficult to keep main results in the main text. Nonetheless, and this is therefore I am keeping this as a minor comment that I hope the authors will find helpful, I suggest the authors attempt to restructure the paper to make key findings more accessible within the main text.

---

> ### Author Rebuttal · Authors · 2025-07-30
>
> We appreciate the reviewer for the constructive comments and for recognizing our contributions as the first to interpret NCO models and to provide a tool that helps reveal the internal mechanisms of these models. This aligns with the core objective of our work, and we are grateful for your recognition. We understand that the main concern lies in the need for broader empirical evaluation, particularly in terms of scalability to larger problem sizes and applicability to diffusion-based models, as well as how to effectively leverage the findings and extend the probing method’s practical implications. We have conducted additional experiments and provided further discussion to address these concerns, as outlined below.
>
> ---
>
> **1. W1-*Experimental setup is limited in scope.***
>
> While we believe analyzing representations on a smaller scale is a meaningful way to isolate complexities and yield insightful findings, we follow your suggestion and include experimental results on TSP-500 and TSP-1000 instances to validate our probing-based approach at larger scales. Since solving large-scale TSPs to optimality is often infeasible, we use approximate solutions as labels instead of exact optima, consistent with prior work. All results are averaged over 1,000 instances.
>
> As shown in the tables below, our key findings persist at larger problem scales. The superiority of LEHD over two heavier encoder-based models becomes even more pronounced in probing results, mirroring its strong performance on large instances and reinforcing our original conclusions. Moreover, LEHD continues to depend on the same key embedding dimensions across scales. This experiment confirms that probing remains an efficient and reliable method for interpreting model behavior in large-scale settings, since it only requires training a lightweight linear probe and thus presents no scalability barrier. We will integrate these results into Table 9 of the manuscript.
>
> | Probing input | *Probing Task 1* | *Probing Task 2* |
> |-----|-----|-----|
> |TSP500|||
> | AM | 0.9240  | 0.71 |
> | POMO |  0.8963  | 0.75 |
> | LEHD |  0.9512   | 0.85 |
> |TSP1000|||
> | AM | 0.9135 | 0.63 |
> | POMO |  0. 8981  | 0.66 |
> | LEHD | 0.9479 | 0.83 |
>
> ---
>
> **2. W2-*Are the behavioral differences due to architectural variations or training algorithm differences?***
>
> Your confusion is completely understandable, as we also encountered this question frequently before conducting this study. It is precisely because it is difficult to pinpoint where differences in model behavior originate that we introduced probing, along with our newly proposed probing methods, to help address this challenge. Below, we first provide a direct answer to your question based on the conclusions from the original manuscript, and then summarize our findings from the manuscript regarding the impact of model architecture and training methods, respectively.
>
> - Directly answer your question: In our original manuscript, we provide an answer to this question: model architecture influences inductive bias, which in turn shapes how models capture decision-relevant information. Models with better generalization tend to encode information using a small number of key dimensions and consistently rely on the same dimensions across different problems. On the other hand, at least in the case of POMO, we find that the training method has relatively little impact on the model's internal behavior. We summarize the relevant findings from our manuscript for your reference below.
>
> - Model architecture: Through the probing analysis, we verify the impact of differences in model architecture. This includes the contrast between heavy encoder models, such as AM and POMO, and heavy decoder models, such as LEHD, in how they capture decision-relevant information. Specifically, LEHD with better generalization tend to consistently utilize the same embedding dimensions across different tasks. In contrast, models whose internal knowledge becomes dispersed or disorganized across dimensions during generalization tend to suffer performance degradation. These findings are discussed in detail in Section 4.3 in page 7 of the manuscript. Furthermore, the additional experiments presented in our response to W3, the details of which are provided below, offer further evidence supporting this conclusion.
>
> - Training methods: We compared different training methods for the POMO model in the manuscript. Using probing, we analyzed the ability of POMO models trained via supervised learning (SL) and reinforcement learning (RL) to capture decision-relevant information. The results show that the SL variant performs slightly worse than its RL counterpart (0.84 vs. 0.86 on Probing Task 2), which is consistent with their overall performance gap (0.571% for SL vs. 0.134% for RL on TSP100). The detailed results can be found in the manuscript, lines 795 to 799.
>
> ---
>
> **3. W3-*The conclusions from CS-Probing lack practical implications and fail to show how they could inform the design of improved NCO methods.***
>
> To address the concern that CS-Probing lacks practical implications, we added **a simple yet effective new experiment** based on the insights derived from CS-Probing in the original manuscript. **Specificly, we conducted an experiment inspired by our probing insight that LEHD generalizes more effectively when its decisions depend on a small subset of embedding dimensions, indicative of sparsity in its final layers. We therefore introduced a regularization term to further promote sparsity in LEHD’s final-layer embeddings, thereby demonstrating how our insights can inform model design.**
>
> Specificlly, we trained LEHD models with varying regularization strengths (λ) on TSP100 and evaluated their generalization on TSP200 and TSP1000. The table reports the performance gap to the best known solutions. The first column (origin, namely λ = 0) corresponds to the unregularized baseline. Moderate regularization (λ = 1e-6 to 1e-3) improves generalization. We will include this experiment to demonstrate the practical value of our probing approach. The current results are preliminary due to time constraints, and a more thorough analysis on more layers/models will be provided in the revised paper.
>
> |     | origin | 1e-6 | 1e-5 | 1e-4 | 1e-3 |
> |-----|-----|-----|-----|-----|-----|
> | TSP100  | 0.57%  | 0.58%  | 0.57%  | 0.57%  | 0.57%  |
> | TSP200 (generalization)   | 0.86%  | 0.88%  | 0.86%  | **0.73%**  | 0.82%  |
> | TSP1000 (generalization)  | 3.17%  | **2.87%**  | 2.93%  | 2.97%  | 3.05%  |
>
> Beyond this, we will enrich our discussions in Section 5 to outline several promising avenues for NCO models to enhance practical implications, including compression, distillation, and pruning of their representation space. Probing NCO layers across CO tasks and training stages may reveal task-specific dimensional requirements, thereby providing a systematic alternative to heuristic selection. We note that while no single study can encompass all possible experiments, our proposed probing pipeline and CS-Probing will serve as powerful tools for future research on the design and analysis of NCO models.
>
> ---
>
> **4. W4&Question-*The benchmarked methods are outdated. Could the proposed CS-probing technique be applied to analyze current Diffusion-based modals as well?***
>
> Yes, probing can also be applied to analyze the current diffusion-based NCO models. For example, the classic model in [1] uses a denoising network that is a GNN, and the node embeddings of the graph in this GNN can be analyzed in the same way as the node embeddings in Transformer-based NCO models for solving routing problems. Additionally, we demonstrate how probing can be applied to GNN models in Section 3.3 on page 5 of the manuscript.
>
> To further address your question, we add a prelimiary experiment of DIFUSCO. Due to version conflicts in DIFUSCO’s code, we rebuilt the environment with alternative packages compatible with our hardware, which prevented us from using its pretrained models. Instead, we present probing results from our own DIFUSCO training (20 epochs), showing that diffusion-based NCO models can also be effectively analyzed with probing.
>
> The table below presents node embedding results for DIFUSCO. As the number of GNN layers increases, the ability to capture Euclidean distances slightly declines, while the ability to identify optimal edges improves. This distinction between low-level and high-level feature encoding aligns with patterns observed in previous models in the manuscript. These results demonstrate that probing is also effective for analyzing diffusion-based models, where the GNN acts as the denoising network.
>
> | Probing input | *Probing Task 1* | *Probing Task 2* |
> |-----|-----|-----|
> | h_init | 0.9476 | 0.49|
> |  h_12 | 0.8710  | 0.73 |
>
> [1] Sun, Zhiqing, and Yiming Yang. "Difusco: Graph-based diffusion solvers for combinatorial optimization."
>
> ---
>
> **5. Minor comment-*The paper relies heavily on the appendix.***
>
> Thank you for this helpful suggestion. We appreciate your understanding of the page limit constraints. We will carefully reread the paper, and if we identify areas where the flow would benefit from additional context, we will incorporate the relevant key results from the appendix into the main text to improve clarity and accessibility.

---

> > ### Comment · Reviewer_veN7 · 2025-08-03
> > **Response**
> >
> > Thanks for the comprehensive rebuttal and new experimental results. It has addressed most of my concerns. I will increase my score accordingly

---

> > > ### Author Response · Authors · 2025-08-03
> > >
> > > We truly appreciate your updated evaluation and constructive feedback. Your comments pushed us to strengthen the experimental section and clarify key aspects, especially in identifying where differences between model behaviors originate. We're glad the new results have addressed your concerns.

---

> > > ### Author Response · Authors · 2025-08-05
> > >
> > > We sincerely thank you again for recognizing our work and rebuttal. We’re especially grateful that you’re considering raising your score from a borderline reject (3) to a positive one. We truly hope this paper can contribute meaningfully to the NCO field, and would greatly appreciate it if you could update the rating at your early convenience.

---

### Official Review · Reviewer_ksMb · 2025-06-30

**Clarity:** 3
**Significance:** 3
**Originality:** 3
**Rating:** 5
**Confidence:** 4

**Summary:**

This paper proposes probing techniques to Neural Combinatorial Optimization (NCO) models, addressing a fundamental gap in the interpretability of these "black-box" systems. The authors investigate the representations learned by SOTA NCO solvers (AM, POMO, LEHD) on the TSP and CVRP. They achieve this through a series of well-designed probing tasks that test for both low-level (e.g., Euclidean distance perception) and high-level (e.g., myopia avoidance) knowledge. The primary methodological contribution is Coefficient Significance Probing (CS-Probing), which analyzes both the magnitude and statistical significance of probe coefficients to identify key embedding dimensions responsible for encoding specific knowledge. The findings reveal that NCO models learn both geometric and strategic information, that different architectures possess distinct inductive biases, and that superior generalization correlates with the consistent reuse of a sparse set of key dimensions.

**Questions:**

My score would increase if the authors provide a stronger comparative context for CS-Probing against other methods or demonstrate the practical utility of their insights by using them to improve a model. And I have the following questions:

1. How does CS-Probing compare, empirically or theoretically, to other interpretability methods like gradient-based attribution or attention visualization for NCO models? What unique insights does it provide? And what is your justification for the sufficiency of linear probes, given that the models themselves are highly non-linear?
2. Have you considered the impact of multiple hypothesis testing when analyzing the significance of 256 dimensions? How might methods to control the false discovery rate (e.g., Benjamini-Hochberg procedure) affect your conclusions about which dimensions are truly important?
3. The finding that two dimensions are sufficient for high performance is striking. How robust is this finding to different initializations or training runs? To establish causality, have you performed ablation studies where you zero out these specific dimensions and measure the performance degradation?
4. Can you provide a concrete example or a proposal for how the insights from CS-Probing could be used to design a better NCO model? For instance, could you design a regularization term that encourages the model to learn sparse, disentangled representations like those you observed in the LEHD model?

**Ethical Concerns:**

["NO or VERY MINOR ethics concerns only"]

**Final Justification:**

Based on these substantial improvements in statistical rigor, practical utility, and empirical validation, I am increasing my score.

**Limitations:**

The authors acknowledge practical limitations in the appendix.

**Quality:**

3

**Strengths And Weaknesses:**

# Strengths:

1. The paper introduces CS-Probing, a novel and fine-grained analytical tool that moves beyond simply detecting the presence of information to identifying where it is encoded. The design of NCO-specific probing tasks is well-motivated and thoughtfully adapted from established practices in NLP and computer vision.
2. This is the first systematic effort to apply probing to NCO models, addressing the critical barrier of model interpretability that currently limits their adoption in high-stakes applications.
3. The experimental evaluation spans multiple models, tasks, and problem scales. The analysis across model layers, training epochs, and generalization scenarios provides a multi-faceted view of model behavior. The finding that just two dimensions can suffice for near-optimal performance is a powerful and actionable insight.
4. The paper is well-written, organized, and effectively uses figures to communicate complex findings.

# Weaknesses:
1. The paper does not provide a theoretical justification for why linear probing is sufficient for analyzing NCO models, nor does it empirically validate this assumption against non-linear probes. Furthermore, the work is presented without comparison to other established interpretability techniques (e.g., gradient-based attribution, attention visualization, neuron ablation), making it difficult to assess the unique advantages of CS-Probing.
2. The analysis involves simultaneously testing 256 dimensions for significance without discussing or applying corrections for multiple hypothesis testing. This raises concerns about the potential for a high false discovery rate, and the statistical validity of the identified key dimensions is not as rigorously established as it could be.
3. The core claims are based primarily on three attention-based models for routing problems. While extensions are mentioned in the appendix, the paper would be far more compelling if it demonstrated the generality of the approach on a fundamentally different architecture (e.g., a GNN) or CO problem in the main text. Additionally, the sensitivity of the findings to the specific design of the probing tasks is not explored.
4. While the insights are analytically interesting, the paper stops short of demonstrating their practical utility. A crucial missing piece is a demonstration of how these interpretability insights could be used to guide the design of a new or improved NCO model, for instance, by applying regularization to encourage sparsity or by building a model that explicitly leverages the identified key dimensions.

---

> ### Author Rebuttal · Authors · 2025-07-30
>
> We thank the reviewer for the constructive and valuable feedback! We appreciate the recognition that our work addresses fundamental gaps, introduces a novel and well-designed framework, overcomes critical barriers, and provides powerful, actionable insights. We also especially thank you for your suggestion to investigate sparsity regularization, which we have found to be beneficial for enhancing generalization. Please find below our responses detailing the new experiments and expanded discussion; we trust these enhancements will further demonstrate the value of our work.
>
> ---
>
> **1. W1&Q1-*Lack justification for using linear probing over non-linear probes and does not compare CS-Probing with other interpretability methods. What unique insights does it provide?***
>
> **I.Why linear probe?**
>
> In representation analysis, linear probing tests whether a feature is encoded in a form that can be extracted by a simple affine transformation. If a property is linearly separable, then the network has organized its internal geometry to make that feature readily available downstream. While NCO models are globally non-linear, our interest lies precisely in whether certain combinatorial features are represented in a geometrically simple (i.e., linearly decodable) form, because the final output layers of NCO models use the embeddings in a linear manner.
>
> For non-linear probes, consider Probing Task 1 (Euclidean distance): although non-linear models may yield better regression performance, it becomes unclear whether the gains stem from the NCO model’s embeddings or the probe’s capacity. Moreover, non-linear probing adds complexity that obscures the internal representations we aim to interpret. Therefore, we adopt linear probing to better align with our goal of understanding how NCO models encode information.
>
> **II. Compare to other interpretability methods**
>
> - Gradient-based attribution. It reveals output sensitivity to input features but does not indicate where or whether specific combinatorial concepts are encoded in hidden representations. It also lacks structural insight into how the model organizes or reasons over the solution space. In contrast, probing offers a systematic interpretability framework by evaluating the linear decodability of target properties from intermediate representations. Our CS-Probing further identifies the specific layers and embedding dimensions where each concept is encoded.
> - Visualization. Attention visualization highlights which nodes or edges the model attends to, but it neither quantifies representational separability nor reveals what specific information is encoded for decision-making in NCO models. In contrast, probing provides a systematic and quantitative assessment of linear separability and information content within embeddings, offering a layer-wise view of representational structure and deeper insight into how decision-relevant information is encoded. Besides, at the beginning of Section 4, we already employ activation visualization, a commonly used interpretability method, as shown in Figure 4. We will also add a discussion of attention visualization in the revised manuscript.
> - Neuron ablation. Eliminating units (by zeroing or randomizing) can help validate the importance of specific neurons, but it is not suitable for identifying which dimensions capture particular information. Zeroing a dimension reveals little about its meaning, and in high-dimensional spaces, it is impractical to test all dimensions or combinations exhaustively. **However, we still include a new experiment based on the neuron ablation method, where we use zeroing to validate the key dimensions identified by CS-Probing. Please refer to the response to Q3 for details.**
>
> **III. Unique insights the CS-Probing provides**
>
> Compared to existing probing methods, CS-Probing offers a deeper level of analysis. As detailed in the manuscript, it not only assesses whether certain information is captured by the deep model's embeddings, but also reveals how that information is encoded. For example, CS-Probing enables us to uncover why LEHD demonstrates better generalization, a capability that standard probing methods typically lack.
>
> **IV. Improvement of the manuscript**
>
> Additional results on neuron ablation (see our response to Q3) and attention visualization will be included in the revised manuscript.
>
> ---
>
> **2. W2&Q2-*No multiple hypothesis testing.***
>
> Thank you for the suggestion. We have incorporated it and added new experiments accordingly. Specifically, we applied the Benjamini-Hochberg procedure to control **the false discovery rate (FDR) at 0.05** across the 256 tested dimensions. We found that the key LEHD dimensions previously identified as significant remain so after correction, **consistent with the results reported in Table 2 of the manuscript**. The updated results and a description of the correction procedure will be included in the revised manuscript.
>
> Notably, the original p-values of these dimensions were well below conventional thresholds (less than 1e-4, as shown in the sample_codes\3_probing_exp\example_probing_exp.ipynb in the anonymous link provided in the manuscript), and they remained significant after FDR correction, further supporting the robustness of our findings.
>
> ---
>
> **3. W3-*Limited evidence of generality across architectures or problems.***
>
> To address your concern, we conducted a preliminary experiment on DIFUSCO [1], a diffusion-based NCO. We trained the model for 20 epochs and then applied probing to analyze its representations. The table below presents node embedding results for DIFUSCO. As the number of GNN layers increases, the ability to capture Euclidean distances slightly declines, while the ability to identify optimal edges improves. This distinction between low-level and high-level feature encoding aligns with patterns observed in previous models in the manuscript. These results demonstrate that probing is also effective for analyzing other architectures, e.g., diffusion-based models, where the GNN acts as the denoising network.
>
> | Probing input | *Probing Task 1* | *Probing Task 2* |
> |-----|-----|-----|
> | h_init | 0.9476 | 0.49|
> |  h_12 | 0.8710  | 0.73 |
>
> [1] Sun, Zhiqing, and Yiming Yang. "Difusco: Graph-based diffusion solvers for combinatorial optimization."
>
> ---
>
> **4. W4&Q4-*Lack a demonstration of how the insights can inform the design of improved NCO models.***
>
> We thank the reviewer for the valuable suggestion on applying regularization to encourage sparsity. We found that, based on CS-probing insights, simply regularizing the final-layer embeddings of the LEHD decoder can directly **improve its generalization**.
>
> In this new experiment, we **simply regularize** the final-layer embeddings of the LEHD decoder to encourage sparsity in its representations, **without making any changes** to the original LEHD model architecture, training hyperparameters, or code implementation.
>
> Specificlly, we trained LEHD models with varying regularization strengths (λ) on TSP100 and evaluated their generalization on TSP200 and TSP1000. The table reports the performance gap to the best known solutions. The first column (origin, namely λ = 0) corresponds to the unregularized baseline. Moderate regularization (λ = 1e-6 to 1e-3) improves generalization. The current results are preliminary due to time constraints, and a more thorough analysis on more layers/models will be provided in the revised paper.
>
> |     | origin | 1e-6 | 1e-5 | 1e-4 | 1e-3 |
> |-----|-----|-----|-----|-----|-----|
> | TSP100  | 0.57%  | 0.58%  | 0.57%  | 0.57%  | 0.57%  |
> | TSP200 (generalization)   | 0.86%  | 0.88%  | 0.86%  | **0.73%**  | 0.82%  |
> | TSP1000 (generalization)  | 3.17%  | **2.87%**  | 2.93%  | 2.97%  | 3.05%  |
>
> We will include this simple yet effective generalization-enhancing experiment in the manuscript to demonstrate the practical value of our proposed probing approach. It provides direct evidence that insights obtained through probing can lead to tangible improvements in model generalization performance.
>
> ---
>
> **5. Q3-*How robust is the two-dimensional finding across runs, and have ablation studies been conducted to test its causal impact (by zero the two key dimensions out) on performance?***
> - **Re-training.** We retrained the LEHD model under different random seeds. We found that although the key dimensions vary across different runs, the conclusion still holds: the LEHD model consistently relies on a small number of key dimensions to make decisions across different scales. Specifically, we find that the retrained LEHD model still relies on two key dimensions to capture decision-relevant information. Although these two dimensions differ from those listed in Table 2 of the manuscript, all of our conclusions remain valid.
> - **Neuron ablation.** We include **an new experiment based on the neuron ablation method**, where we use zeroing to validate the key dimensions identified by CS-Probing. As shown in the table below, zeroing LEHD’s **key dimensions (31 and 97)** results in a significant performance drop (over 60%). Even zeroing either one individually causes a larger gap than zeroing other dimensions. For comparison, we zeroed two non-key dimensions: 98 (with large activation in Figure 4 of the manuscript) and 126 (with the highest probing coefficient among non-key dimensions in Table 2). In addition to these new experimental results, the original results in Table 3 of the manuscript also show that LEHD continues to rely on the same two key dimensions when generalizing to larger problem sizes. These results further confirm the importance of the key dimensions identified by CS-Probing.
>
> | Zeroing Dimension(s)     | Gap |
> |--------|--------|
> | None(original) | 0.57% |
> | 31, 97 | 60.43% |
> | 31 | 0.90% |
> | 97 | 1.80% |
> | 98 | 0.59% |
> | 126 | 0.60% |
> | 98, 126 | 0.62% |
>
> We will include this new experiment and observation in the updated manuscript to enhance the robustness of our findings.

---

> > ### Comment · Reviewer_ksMb · 2025-08-03
> >
> > Thank you for the comprehensive and convincing rebuttal. Your responses effectively address my key concerns. Based on these substantial improvements in statistical rigor, practical utility, and empirical validation, I am increasing my score.

---

> > > ### Author Response · Authors · 2025-08-03
> > >
> > > Thank you very much for your kind feedback and for increasing your score. We truly appreciate your initial recognition and valuable suggestions, as well as your acknowledgment of our efforts to improve the statistical rigor, practical value, and empirical validation. Your comments, especially your suggestions on practical utility, were instrumental in helping us strengthen the paper.

---

### Official Review · Reviewer_LZ81 · 2025-07-02

**Clarity:** 2
**Significance:** 2
**Originality:** 3
**Rating:** 4
**Confidence:** 2

**Summary:**

This paper proposes Coefficient Significance Probing (CS-Probing) to enable deeper analysis of representations for neural combinatorial models. Existing neural combinatorial models are black-box models that lack interpretability. CS-Probing reveals that different NCO models introduce diverse inductive biases into the representations. The authors also try to uncover the mechanisms of generalization in NCO models.

**Questions:**

1. Diffusion models have shown good performance in combinatorial optimizations. What about the analysis in diffusion models? Do the discoveries also hold in diffusion-based solvers?

2. LEHD uses key dimensions for decision-making. Although the optimality gap increase as we use the two-dimension embeddings. Can we use less embeddings, such as 8, 16, 32 or 64 dimensions, while the performance does not decrease?

**Ethical Concerns:**

["NO or VERY MINOR ethics concerns only"]

**Final Justification:**

After reviewing the detailed response, I now understand the significance and technical challenges of integrating interpretability methods into the NCO domain. The novelty of the paper has been clarified, and the additional experiment has strengthened my confidence in the validity of the claims. I believe the authors have sufficiently addressed my concerns, and I trust that my primary feedback will be incorporated as promised.

Regarding the camera-ready version, I recommend including the added experiments (such as the TSPLib experiment) in the main paper or at least in the appendix. Also, as mentioned in the author's response, a thorough discussion of further exploration of probing for diffusion-based models would further enhance the paper. I recommend to add it as well, e.g., in the conclusion part as a future work direction.

Under these conditions, I could support a score of 4.

**Limitations:**

The paper does not propose an effective algorithm to enhance the representation learning or generalization of the neural combinatorial solvers.

**Quality:**

3

**Strengths And Weaknesses:**

Strengths:

1. The paper proposes a method to interpret the black-box neural combinatorial models. The task is meaningful in the field of combinatorial optimization.
2. The paper offers insights into the representations, inductive biases and generalization mechanisms of the neural combinatorial models.

Weaknesses:
1. The authors focus on analyzing the representations of the neural combinatorial models. However, I encourage the authors to provides some representation learning methods to improve the performance and generalization ability of combinatorial solvers based on the discoveries.
2. I find the paper uses instances with only 20, 100 and 200 nodes in the experiments. I wonder whether the results also hold in large scale combinatorial problems. In fact, we are more interested in the analysis and improvement in the large-scale problems.
3. The details and the usage of linear probing model is not clear. The authors may explain the probing process.

---

> ### Author Rebuttal · Authors · 2025-07-30
>
> We appreciate the reviewer’s constructive feedback and recognition of our contributions to interpreting NCO models, particularly our insights into representation learning, inductive biases, and generalization mechanisms. This was the core motivation of our study, and we appreciate your recognition. We understand the main concern is regarding how to leverage these findings and the broader probing applicabilitys, which we address with new experiments and discussions as follows.
>
> ---
>
> **1. W1&Limitations-*The authors focus on analyzing representations in neural combinatorial models. I encourage them to propose representation learning methods to enhance solver performance and generalization.***
>
> Thanks for the suggestion! Before we provide new experiments to directly show how to translate our findings into actionable designs, we would like to first emphasize that:
>
> * **We introduce the first probing framework for NCO and offers the first comprehensive study of internal representations in existing NCO models.** We regard our work as a meaningful and necessary step, as our interpretability framework can provide the foundation for more principled novel designs in the future.
> * **More importantly, we enhances transparency and address the key questions raised by the operations research (OR) community.** We uncover what knowledge NCO models capture and how they are leveraged to solve CO problems. These represent foundational questions raised by the OR community concerning the soundness of NCO methods. Our work directly studies them: NCO models capture both low-level features such as Euclidean distance and high-level strategies like avoiding myopic decisions.
> * **We provide new analytical tools (CS-Probing).** Through CS-Probing, we find that: (1) distinct inductive biases learned by different NCO models; (2) they encode decision rationales in specific embedding dimensions closely tied to generalization; and (3) in the subspace formed by these key dimensions, model decisions can be interpreted by examining only a few informative dimensions.
>
> We believe our findings are worth sharing with the community, which aligns well with many related interpretability research in deep learning. For example: (1) In computer vision, [1] employs probing to evaluate CNN layer representations, revealing that deeper layers produce increasingly linearly separable features, which informs future representation evolution, skip connection design, etc; (2) In natural language processing, [2] demonstrates that probing enhances interpretability, guides architecture design, uncovers latent capabilities, and inspires representation manipulation and alignment.
>
> **We agree with the reviewer that designing methods based on our findings is exciting. To show such potential, we conducted an experiment inspired by our probing insight that LEHD generalizes more effectively when its decisions depend on a small subset of embedding dimensions, indicative of sparsity in its final layers. We therefore introduced a regularization term to further promote sparsity in LEHD’s final-layer embeddings, thereby demonstrating how our insights can inform model design.**
>
> Specificlly, we trained LEHD models with varying regularization strengths (λ) on TSP100 and evaluated their generalization on TSP200 and TSP1000. The table reports the performance gap to the best known solutions. The first column (origin, namely λ = 0) corresponds to the unregularized baseline. Moderate regularization (λ = 1e-6 to 1e-3) improves generalization. We will include this experiment to demonstrate the practical value of our probing approach. The current results are preliminary due to time constraints, and a more thorough analysis on more layers/models will be provided in the revised paper.
>
> |     | origin | 1e-6 | 1e-5 | 1e-4 | 1e-3 |
> |-----|-----|-----|-----|-----|-----|
> | TSP100  | 0.57%  | 0.58%  | 0.57%  | 0.57%  | 0.57%  |
> | TSP200 (generalization)   | 0.86%  | 0.88%  | 0.86%  | **0.73%**  | 0.82%  |
> | TSP1000 (generalization)  | 3.17%  | **2.87%**  | 2.93%  | 2.97%  | 3.05%  |
>
> Beyond this, we will enrich our discussions in Section 5 to outline several promising avenues for enhancing NCO models, including compression, distillation, and pruning of their representation space. Probing NCO layers across CO tasks and training stages may reveal task-specific dimensional requirements, thereby providing a systematic alternative to heuristic selection. We note that while no single study can encompass all possible experiments, our proposed probing pipeline and CS-Probing will serve as powerful tools for future research.
>
> [1] Alain, Guillaume, and Yoshua Bengio. "Understanding intermediate layers using linear classifier probes."
>
> [2] Allen-Zhu, Zeyuan, and Yuanzhi Li. "Physics of language models: Part 3.1, knowledge storage and extraction."
>
> ---
>
> **2. W2-*Large scale problems.***
>
> While we believe analyzing representations on a smaller scale is a meaningful way to isolate complexities and yield insightful findings, we follow your suggestion and include experimental results on TSP-500 and TSP-1000 instances to validate our probing-based approach at larger scales. Since solving large-scale TSPs to optimality is often infeasible, we use approximate solutions as labels instead of exact optima, consistent with prior work. All results are averaged over 1,000 instances.
>
> As shown in the tables below, our key findings persist at larger problem scales. The superiority of LEHD over two heavier encoder-based models becomes even more pronounced in probing results, mirroring its strong performance on large instances and reinforcing our original conclusions. Moreover, LEHD continues to depend on the same key embedding dimensions across scales. This experiment confirms that probing remains an efficient and reliable method for interpreting model behavior in large-scale settings, since it only requires training a lightweight linear probe and thus presents no scalability barrier. We will integrate these results into Table 9 of the manuscript.
>
> | Probing input | *Probing Task 1* | *Probing Task 2* |
> |-----|-----|-----|
> |TSP500|||
> | AM | 0.9240  | 0.71 |
> | POMO |  0.8963  | 0.75 |
> | LEHD |  0.9512   | 0.85 |
> |TSP1000|||
> | AM | 0.9135 | 0.63 |
> | POMO |  0. 8981  | 0.66 |
> | LEHD | 0.9479 | 0.83 |
>
> ---
>
> **3. W3-*The linear probing model and probing process are unclear.***
>
> The linear probing model is a linear model, specifically
>
> $y=\beta_0+\beta_1 x_1 + ... \beta_n x_n$
>
> where $y$ represents the information to be probed, $x_n$ is the activation value of the $n$-th dimension of the embedding of the NCO model, and $\beta$ is the regression coefficient.
>
> The probing process is detailed in Section 3.1 and Appendix B. Briefly, probing tests whether a deep model encodes specific information by training a simple (typically linear) model to predict target labels from the model’s internal activations. Successful prediction indicates that the relevant information is present in the embeddings. This approach is standard in other domains as well.
>
> We will revise the paper to more clearly describe the linear probing model. Please let us know if there are any specific points of confusion that we can further clarify.
>
> ---
>
> **4. Q1-*Do the probing analyses also apply to diffusion-based models?***
>
> Yes, probing can also be applied to analyze diffusion-based NCO models. For example, the classic model in [1] uses a denoising network that is a GNN, and the node embeddings of the graph in this GNN can be analyzed in the same way as the node embeddings in Transformer-based NCO models for solving routing problems. Additionally, we demonstrate how probing can be applied to GNN models in Section 3.3 on page 5 of the manuscript.
>
> To further address your question, we conducted a preliminary experiment on DIFUSCO. We trained the model for 20 epochs and then applied probing to analyze its representations. The table below presents node embedding results for DIFUSCO. As the number of GNN layers increases, the ability to capture Euclidean distances slightly declines, while the ability to identify optimal edges improves. This distinction between low-level and high-level feature encoding aligns with patterns observed in previous models in the manuscript. These results demonstrate that probing is also effective for analyzing diffusion-based models, where the GNN acts as the denoising network.
>
> | Probing input | *Probing Task 1* | *Probing Task 2* |
> |-----|-----|-----|
> | h_init | 0.9476 | 0.49|
> |  h_12 | 0.8710  | 0.73 |
>
> [1] Sun, Zhiqing, and Yiming Yang. "Difusco: Graph-based diffusion solvers for combinatorial optimization."
>
> ---
>
> **5. Q2-*Can we use less embeddings for LEHD, while the performance does not decrease?***
>
> Insightful questions! Dimensionality selection in deep learning models is a complex and interesting problem. To first answer the question: the LEHD model as a whole does not achieve comparable performance with very low-dimensional embeddings (e.g., 8 or 16 dimensions). Although the LEHD model appears to rely primarily on just two dimensions of the final-layer embeddings to make near-equivalent decisions, this does not imply that the entire model operates using only these two dimensions. The earlier layers still require high-dimensional representations to effectively learn and process information.
>
> We agree with the reviewer that developing a principled method for determining the necessary dimensions of NCO models while retaining perforamnce is an exciting future work. Notably, the additional experiments in our response to W1 have already provided preliminary evidence that insights from our probing analysis can improve LEHD’s generalization performance while employing fewer active embeddings via sparsity regularization. We consider these findings both compelling and worthy of extension to other layers in future work, where our proposed probing pipeline/tools will furnish a systematic approach for identifyin key dimensions of each layer.

---

> > ### Comment · Reviewer_LZ81 · 2025-08-04
> >
> > Thanks for your feedback! However, I still have some concerns regarding about the novelty, experimental results, and practical relevance of this paper. My concerns are outlined below:
> >
> > - First, DIFUSCO  is proposed several years ago. However, both the model architecture and the reported results of DIFUSCO are already outdated relative to the current SOTA. The authors attribute performance improvements to their proposed method, but fail to isolate the root cause of such gains. Without a thorough ablation study or a direct comparison to stronger and more recent diffusion-based baselines (e.g., T2T, FastT2T, or optimization-consistent variants), it is difficult to conclude whether the improvements truly arise from methodological innovation or from superficial design or training changes. I recommend the authors to evaluate their method against stronger, more relevant diffusion-based baselines. Otherwise, the claims of performance improvement appear unconvincing and potentially misleading.
> > - Second, the authors draw general conclusions about the behavior of diffusion models based on their empirical results. However, these conclusions seem insufficiently supported by broader empirical trends. For example, prior studies suggest that deeper diffusion architectures often lead to better performance, while reducing layers may degrade solution quality. This contradicts some of the claims or implicit assumptions in the paper. Without a more comprehensive analysis of how diffusion depth, noise scheduling, or sampling strategy interact with the task performance, it is risky to generalize the observations. I would encourage the authors to revisit these conclusions, and either provide more controlled experiments or significantly qualify the scope of their claims.
> > - Finally, although the authors added some larger-scale results (e.g., on TSP1000), the overall experimental scale still remains limited as it seems. More critically, the method is not evaluated on any real-world benchmark datasets, such as TSPLIB, which is a standard testbed for generalization in the CO field. This significantly undermines the credibility of their method’s generalization ability. In particular, I would like to see more insights into why the method should perform well under out-of-distribution settings like TSPLIB. As it stands, there is little evidence or theoretical justification for robustness beyond synthetic distributions. In fact, based on my understanding of the method's design, I am skeptical that the proposed method will generalize well to unseen real-world problem structures.
> >
> >  [1] From Distribution Learning in Training to Gradient Search in Testing for Combinatorial Optimization
> >  [2] Fast T2T: Optimization Consistency Speeds Up Diffusion-Based Training-to-Testing Solving for Combinatorial Optimization
> >  [3] Unify ML4TSP: Drawing Methodological Principles for TSP and Beyond from Streamlined Design Space of Learning and Search
> >  [4] COExpander: Adaptive Solution Expansion for Combinatorial Optimization

---

> > > ### Author Response · Authors · 2025-08-05
> > > **Response to your concerns and clarify of misunderstandings (1/4)**
> > >
> > > Thank you for your response and your willingness to engage in the discussion. **However, some of your latest comments are somewhat confusing to us, as they appear to reflect major misunderstandings and may partially due to conflating our work with content from other papers.**  We value your detailed feedback and have incorporated additional discussions and experiments. We hope the clarifications below will help address these potential issues and your remaining concerns.
> > >
> > >
> > > > Thanks for your feedback! However, I still have some concerns regarding about the novelty, experimental results, and practical relevance of this paper.
> > >
> > > Thank you for your reply. We hope our previous rebuttal has adequately addressed your concerns regarding how to leverage our probing-based findings to inform representation learning methods aimed at improving the performance of NCO solvers.
> > >
> > > Regarding novelty, this is a new comment, and we would like to clarify that our work is the first comprehensive study leveraging probing to analyze the representations of NCO models. **As you noted in your initial review and from all other reviewers,
> > >  we propose the first systematic approach to interpret the black-box NCO models, which makes a meaningful contribution to the field and offers insights into the representations, inductive biases, and generalization mechanisms of the NCO models.**
> > >
> > >
> > > Regarding the experimental results and the practical relevance of this paper, we believe you are referring to the following specific points, to which we respond as follows.
> > >
> > > > First, DIFUSCO is proposed several years ago. However, both the model architecture and the reported results of DIFUSCO are already outdated relative to the current SOTA. The authors attribute performance improvements to their proposed method, but fail to isolate the root cause of such gains. Without a thorough ablation study or a direct comparison to stronger and more recent diffusion-based baselines (e.g., T2T, FastT2T, or optimization-consistent variants), it is difficult to conclude whether the improvements truly arise from methodological innovation or from superficial design or training changes. I recommend the authors to evaluate their method against stronger, more relevant diffusion-based baselines. Otherwise, the claims of performance improvement appear unconvincing and potentially misleading.
> > >
> > > **We are deeply confused by these comments. They seem to have potentially conflated with other papers you review. We would like to clarify a major misunderstanding here: our work does not claim to propose a new diffusion-based method. Instead, our paper is a work at the line of interpretability reserach, to systematically investigate whether probing can be used to analyze the representation learned by NCO models, rather than proposing or improving a specific NCO model.** During the rebuttal process, we add DIFUSCO specifically to address your question in Q1: “Do the discoveries also hold in diffusion-based solvers?”. However, this does not change our main focuse, we are not introducing a new diffusion-based method, nor aiming to outperform any difussion-based baselines. We sincerely apologize if this was the impression given. If you could kindly point out where this confusion arose, we will be more than happy to revise the manuscript to clarify our intentions.
> > >
> > > We also provide the following discussions:
> > >
> > > - If by “performance improvements” you are referring to the enhancements observed in LEHD, we would like to clarify that Sections 4.2 to 4.4 of our manuscript (pages 6–8) are devoted to a step-by-step analysis and experimental investigation that identifies the "root cause" of these improvements. Specifically, they explain why applying regularization to the final-layer embeddings of LEHD is effective. This insight directly aligns with the suggestion raised by Reviewer ksMb. The new experiments we added in the rebuttal serve to validate the conclusions derived from our probing-based analysis, rather than presenting a newly proposed enhancement method aimed at achieving SOTA performance.
> > > - If by "performance improvements" you are referring to DIFUSCO, we would like to clarify that we never claimed to improve DIFUSCO's performance in our rebuttal. Our preliminary exploration of DIFUSCO was solely intended to demonstrate that, like transformer-based approaches, it can also be effectively analyzed using probing techniques. This (together with the Section 3.3 in manuscript) supports our broader claim that probing serves as a general tool for analyzing a wide range of NCO models, regardless of architecture.
> > >
> > > Once again, we are unsure which part of the manuscript or rebuttal responses may have led to this misunderstanding. If you could kindly point it out, we would be more than happy to revise it accordingly.

---

> > > ### Author Response · Authors · 2025-08-05
> > > **Response to your concerns and clarify of misunderstandings (2/4)**
> > >
> > > > Without a thorough ablation study or a direct comparison to stronger and more recent diffusion-based baselines, it is difficult to conclude whether the improvements truly arise from methodological innovation or from superficial design or training changes. I recommend the authors to evaluate their method against stronger, more relevant diffusion-based baselines. Otherwise, the claims of performance improvement appear unconvincing and potentially misleading.
> > >
> > > Thank you for providing these references. While our primary focus is **NOT** to propose new diffusion models, we appreciate your suggestion to test our probing pipeline on diffusion-based models to demonstrate the broader applicability of our pipeline. We have made an effort to gather some preliminary probing results for a DIFUSCO model in our early response. However, due to time limits of rebuttal, we acknowledge that our preliminary results are not sufficient to draw definitive conclusions for all the mentioned diffusion models. We believe that such an in-depth investigation, including probing-based insights and analysis using our proposed CS-Probing method, would be valuable enough to constitute a standalone future work. Nevertheless, we commit to further exploring probing for these mentioned models in the cam-ready version, including citing these papers and adding the corresponding results and providing a thorough discussion of all the referenced works.
> > >
> > > **In the following, we explain why prioritizing representative constructive neural solvers that address CO in an end-to-end manner, rather than diffusion-based models that are NOT fully end-to-end, is more meaningful for this work:**
> > >
> > > -  **The included methods better reflect end-to-end decision-making processes in NCO**. Since our work is the first to introduce interpretability methods into the NCO domain, we focused on **end-to-end NCO models** (i.e., **constructive NCO models**) in the manuscript to analyze their internal representations. This choice was made to avoid confounding factors beyond the learned representations. In contrast, models like heatmap-based or improvement NCO methods often depend on external search strategies, which may obscure the direct impact of the learned representations.
> > > - **We have already analyzed a sufficient number of representitive models**. We explored the most representative transformer-based models, including heavy encoder architectures such as AM and POMO, as well as the heavy decoder model LEHD; we examined how classical models like MatNet encode edge information in non-Euclidean spaces; and we investigated whether the embeddings of GNN-based models can be effectively probed; we selected DIFUSCO as a representative case to show that probing can also be applied to diffusion-based models. For the diffusion-based NCO, we hope this preliminary result sufficiently **demonstrates feasibility**, while leaving more comprehensive probing analyses of diffusion-based models to future studies.
> > > - **We have already analyzed a sufficient number of representitive CO problems** We tested a diverse range of combinatorial problems, including TSP, CVRP, ATSP, and JSSP; across different instance sizes from small to large; across different data distributions, from uniform to realistic ones (see TSPLib results below); and across different levels of decision-related features, from low-level to high-level.
> > > - **Our choice of models aligns with established practices in the probing literature**. This paper is the first to apply probing in the NCO domain and to propose a novel method for uncovering internal mechanisms of NCO models. Our aim is not to conduct exhaustive benchmarking by standardizing datasets and running numerous models, as is typical in benchmark-track studies. As discussed in our Related Work section, similar precedents exist in other fields. For instance, an ICLR 2024 paper [1] used probing to explore whether LLM representations reflect real-world knowledge, analyzing only LLaMA and Pythia despite the many available LLMs. Likewise, a NeurIPS 2023 paper [2] probed just two classical LLMs to study in-context learning. These examples show that early-stage exploratory work focuses on generating insights into model behavior, not exhaustive evaluation. Our approach follows this same philosophy.
> > >
> > > We believe that once it is clear our work focuses on using probing to analyze NCO models rather than developing new SOTA models, concerns about the lack of performance comparisons on TSP/CVRP with SOTA methods should be alleviated. To further address your comments on "baselines," we have included additional interpretability techniques in the rebuttal as a point of comparison to probing, further enriching our study. Please see our response to Reviewer ksMb under 'W1&Q1' for details.
> > >
> > > [1] Gurnee, Wes, and Max Tegmark. "Language models represent space and time."
> > >
> > > [2] Zhao, Siyan, et al. "Probing the decision boundaries of in-context learning in large language models."

---

> > > ### Author Response · Authors · 2025-08-05
> > > **Response to your concerns and clarify of misunderstandings (3/4)**
> > >
> > > > Second, the authors draw general conclusions about the behavior of diffusion models based on their empirical results. However, these conclusions seem insufficiently supported by broader empirical trends. For example, prior studies suggest that deeper diffusion architectures often lead to better performance, while reducing layers may degrade solution quality. This contradicts some of the claims or implicit assumptions in the paper.
> > >
> > > Regarding this concern, we believe that our response to the first point has already addressed most of the issues raised here. Therefore, we will not repeat those explanations, but will instead focus on clarifying one potential misunderstanding.
> > >
> > > **Regarding "*this contradicts some of the claims or implicit assumptions in the paper*", our preliminary findings are fully aligned with your conclusion "*deeper diffusion architectures often lead to better performance, while reducing layers may degrade solution quality*".** As shown in the table in resoonse to Q1, for *Probing Task 2*, which evaluates the model’s ability to capture high-level decision features such as avoiding myopic selections, shows that DIFUSCO exhibits clear improvement as depth increases. The AUC rises from 0 in the early layers to 0.73 in deeper layers, indicating stronger decision-awareness. In contrast, for low-level features such as Euclidean distance awareness (i.e., *Probing Task 1*), the initial layers demonstrate the strongest signals due to the sinusoidal encoding of raw coordinates. As the network deepens, its embeddings increasingly encode more abstract and high-level features, which may slightly reduce sensitivity to raw distance. This trend is consistent with what we observed in transformer-based models in the manuscript (see Figure 2 of the manuscript).
> > >
> > > Morever, we clarify that "Although the LEHD model appears to rely primarily on just two dimensions of the final-layer embeddings to make near-equivalent decisions, this does not imply that the entire model operates using only these two dimensions. The earlier layers still require high-dimensional representations to effectively learn and process information." Our intention is to suggest that models can potentially be guided to learn more compact representations, for example, by regulating the decision layers as shown in our added experiments. In the future, this could also be extended to intelligently determining the embedding dimensionality at different layers to improve generalization, which we leave as an exciting future work. If our previous response to Q1/Q2 caused any misunderstanding, we sincerely apologize and are happy to provide further clarification.
> > >
> > > >  Without a more comprehensive analysis of how diffusion depth, noise scheduling, or sampling strategy interact with the task performance, it is risky to generalize the observations. I would encourage the authors to revisit these conclusions, and either provide more controlled experiments or significantly qualify the scope of their claims.
> > >
> > > Thank you for highlighting comprehensive analysis of diffusion depth, noise scheduling, and sampling strategies in diffusion-based models. As discussed earlier, diffusion-based models were not prioritized in our work because they are not fully end-to-end, although we recognize this as an exciting direction for future research. Due to the time constraints of the rebuttal phase, we acknowledge that our preliminary results are insufficient to draw definitive conclusions for all the mentioned diffusion models.
> > >
> > > Nevertheless, we believe that all of the provided four papers are amenable to probing analysis, as they are based on GNN architectures. This is supported by the experiments in Section 3.3 of the manuscript and the additional experiments of DIFUSCO presented in the rebuttal, which demonstrate that probing can analyze such models. We will cite these works in the updated version of the manuscript and explicitly discuss the feasibility of applying probing to analyze them. We hope that future research will build upon our probing techniques and the perspectives you outlined to uncover the internal mechanisms of diffusion-based NCO models and potentially yield actionable insights or even models with improved performance. **We believe that such an in-depth investigation with our probing pipeline would be valuable enough to constitute a standalone future work, ideally conducted in collaboration with experts like you to deepen the understanding of diffusion models for NCO.**

---

> > > ### Author Response · Authors · 2025-08-05
> > > **Response to your concerns and clarify of misunderstandings (4/4)**
> > >
> > > > Finally, although the authors added some larger-scale results (e.g., on TSP1000), the overall experimental scale still remains limited as it seems.
> > >
> > > As emphasized, our goal is not to propose a new NCO method for large-scale instances, but to uncover insights into why NCO solvers work for CO problems. We believe the studied scales are sufficient for our research objectives. Moreover, we included results on TSP1000, which is already considered large. In practice, real-world TSP and CVRP instances typically have fewer than 1,000 nodes due to operational constraints.
> > >
> > > Moreover, as summarized above in response to your second point, we have conducted extensive experiments to demonstrate the utility of probing for analyzing NCO models. **Reviewer Mbie acknowledged this by noting that "The study is extensive and provides a large number of experiments and data." The other two reviewers also described our experimental analysis as “comprehensive” during the rebuttal period**.
> > >
> > > Lastly, we fully acknowledge that the adequacy of experiments in a single paper is ultimately a subjective judgment. To further address your concern, we have conducted additional experiments based on your suggestions, as detailed below.
> > >
> > > > More critically, the method is not evaluated on any real-world benchmark datasets, such as TSPLIB, which is a standard testbed for generalization in the CO field. This significantly undermines the credibility of their method’s generalization ability.
> > >
> > > Thank you for your suggestion. In response, we conducted additional experiments to show that insights from probing can improve LEHD's generalization not only to larger problem sizes but also across data distributions—specifically, models trained on uniform instances and tested on TSPLib. The last row of the table below shows that adding a regularization term to encourage sparsity in LEHD’s final-layer embeddings **improves generalization on TSPLib**, demonstrating how our insights can inform model design.
> > >
> > > ||origin|1e-6|1e-5|1e-4|1e-3|
> > > |-----|-----|-----|-----|-----|-----|
> > > |TSP100|0.57%|0.58%|0.57%|0.57%|0.57%|
> > > |...||||||
> > > |...||||||
> > > |TSPLib(generalization)|  5.26%  | **4.94%**  | 4.99%  | 5.05%  | 8.61%|
> > >
> > > **It is important to note that we have never claimed, nor intended to demonstrate, that the modified LEHD model is a SOTA model in the NCO field. Instead, this example serves as a preliminary experimental example to illustrate how probing, particularly our proposed CS-Probing method, can be used to investigate the internal mechanisms of NCO models and derive actionable insights that inform future model exploration and design.**
> > >
> > > > In particular, I would like to see more insights into why the method should perform well under out-of-distribution settings like TSPLIB. As it stands, there is little evidence or theoretical justification for robustness beyond synthetic distributions.
> > >
> > > Thanks for the suggestion! This is precisely the question that motivates our work: Why do black-box NCO models work? Why can they generalize? Why do some models generalize better than others? For instance, the original LEHD paper shows superior generalization to TSPLib compared to two baselines (Table 2, page 8 of LEHD paper), but it does not explain why. In our manuscript (Sections 4.2 to 4.4, pages 6–8), we address this gap using probing, particularly our proposed CS-Probing method. Through a step-by-step analysis, we find that **different model architectures induce different inductive biases**. Compared to heavy-encoder models like AM and POMO, LEHD **encodes decision-relevant information into a small number of key embedding dimensions and consistently relies on these dimensions when solving problems across different scales and distributions** (including TSPLib).
> > >
> > > We believe the probing approach opens the door for future work to demystify other NCO models and systematically answer questions like the one you raised.
> > >
> > > >  In fact, based on my understanding of the method's design, I am skeptical that the proposed method will generalize well to unseen real-world problem structures.
> > >
> > > - If “proposed method” refers to our modified LEHD, we note that while our additional experiments (see above) provide preliminary evidence of improved generalization, **the goal is not to propose a new SOTA NCO model**, but to analyze existing ones and show how probing reveals actionable insights—already supported by the modified LEHD's improved performance to realistic instances.
> > > - If “proposed method” refers to the probing methodology, we clarify that it applies to any TSP/CVRP instance distribution. **Probing operates on the NCO model’s numerical embeddings, regardless of whether inputs are synthetic or real-world**. **Its goal is to analyze internal representations rather than improve performance, and the probing model's accuracy serves as a diagnostic metric**. As long as embeddings are available, the analysis remains valid. For details, please refer to our response to W3 in the first-round rebuttal.

---

> > > > ### Comment · Reviewer_LZ81 · 2025-08-05
> > > >
> > > > After reviewing the detailed response, I now better understand the significance and technical challenges of integrating interpretability methods into the NCO domain. The novelty of the paper has been clarified, and the additional experiment has strengthened my confidence in the validity of the claims. I believe the authors have sufficiently addressed my concerns, and I trust that my primary feedback will be incorporated as promised.
> > > >
> > > > Regarding the camera-ready version, I recommend including the added experiments (such as the TSPLib experiment) in the main paper or at least in the appendix. Also, as mentioned in your response, a thorough discussion of further exploration of probing for diffusion-based models would further enhance the paper. I recommend to add it as well, e.g., in the conclusion part as a future work direction.
> > > >
> > > > Under these conditions, I could support a score of 4.

---

> > > > > ### Author Response · Authors · 2025-08-05
> > > > >
> > > > > We sincerely thank the reviewer for supporting our work and recommend acceptance. We are pleased that our discussions have sufficiently addressed your concerns. Once again, thank you for your willingness to engage in the discussion and for your valuable time and constructive feedback. Your comments have been instrumental in shaping and strengthening our paper.
> > > > >
> > > > > We promise we will incorporate all discussion points from you and other reviewers into the camera-ready revision. We fully agree that the Conclusion should include a discussion of future work on applying our probing pipeline to recently developed diffusion-based NCO models; specifically, we will cover the four models [1–4] you mentioned and explicitly highlight how probing can illuminate their underlying mechanisms as we discussed. In addition, we will add the TSPLib results at the end of Section 4.4 as further validation of the actionable insights derived from our CS-Probing analysis.
> > > > >
> > > > > We believe that our work represents the first systematic effort to interpret black-box NCO models, showcasing probing as a valuable tool for analyzing their internal mechanisms and revealing impactful insights for the NCO community. We will further organize the GitHub repository linked anonymously in the manuscript and make it public in the camera-ready version to better support future research efforts in this direction.
> > > > >
> > > > >
> > > > >
> > > > > [1] T2T: From distribution learning in training to gradient search in testing for combinatorial optimization. NeurIPS 2023
> > > > >
> > > > > [2] Fast T2T: Optimization consistency speeds up diffusion-based training-to-testing solving for combinatorial optimization." NeurIPS 2024
> > > > >
> > > > > [3] Unify ML4TSP: Drawing methodological principles for tsp and beyond from streamlined design space of learning and search. ICLR 2025
> > > > >
> > > > > [4] COExpander: Adaptive Solution Expansion for Combinatorial Optimization. ICML 2025

---

### Official Review · Reviewer_MBie · 2025-07-03

**Clarity:** 2
**Significance:** 2
**Originality:** 3
**Rating:** 5
**Confidence:** 3

**Summary:**

The paper investigates internal representations of attention-based models used for combinatorial optimisation (AM, POMO, LEHD). It does it in two ways: a. by using internal representations to solve auxiliary tasks b. by looking at significant dimensions of these vectors utilised by the models.

**Questions:**

Q1. One of the outcomes of your study is that CS-probing can identify the most significant dimensions, which helps to build a smaller model with similar performance. However, the same outcome can be achieved by using hyperparameter tuning methods. What are the other practical implications you envision?

Q2. The second task focuses on the models ability to identify parts of the optimal solution. However, it isn't always unique. Moreover, there might be a drastically different solution with very similar optimisation cost. How do address this issue in your study?

**Ethical Concerns:**

["NO or VERY MINOR ethics concerns only"]

**Final Justification:**

The authors have addressed my concerns and I recommend acceptance.

**Limitations:**

yes

**Quality:**

3

**Strengths And Weaknesses:**

Pros:
The use of probing for optimisation models is interesting, since it can identify limitations and potential improvement opportunities for existing models. It was interesting to see myopia avoidance and very small number of significant dimensions in the internal representations.

The study is extensive and provides a large number of experiments and data

Cons:
Large parts of the paper contained detailed discussions of internal embeddings (e.g. these are significant dimensions or the models are not purely greedy optimisers) rather than actionable conclusions. Table 3 is an example when this further step was taken based on CS-probing at least to reduce the model size, but the general probing idea with auxiliary tasks has unclear actionable implications.

---

> ### Author Rebuttal · Authors · 2025-07-30
>
> We appreciate the reviewer for the constructive feedback and for recognizing the importance of our interpretability exploration in NCO models, as well as the interesting and valuable findings we present. We are also grateful for the acknowledgment that our study is extensive and supported by a large number of experiments and data. The probing methodology and the dataset we constructed reflect the core contributions we hope to bring to the NCO field, and we sincerely appreciate your recognition of their value. We understand that the main concern lies in the lack of actionable implications of the probing results in the current manuscript, which we address through new experiments and additional discussions as outlined below.
>
> ---
>
> **1. Cons-*The probing idea with auxiliary tasks has unclear actionable implications.***
>
> **To address the concern that CS-Probing lacks actionable implications, we added a simple yet effective new experiment based on the insights derived from CS-Probing in the original manuscript.** Specificly, we conducted an experiment inspired by our probing insight that LEHD generalizes more effectively when its decisions depend on a small subset of embedding dimensions, indicative of sparsity in its final layers. We therefore introduced a regularization term to further promote sparsity in LEHD’s final-layer embeddings, thereby demonstrating how our insights can inform model design.
>
> Specificlly, we trained LEHD models with varying regularization strengths (λ) on TSP100 and evaluated their generalization on TSP200 and TSP1000. The table reports the performance gap to the best known solutions. The first column (origin, namely λ = 0) corresponds to the unregularized baseline. Moderate regularization (λ = 1e-6 to 1e-3) improves generalization. We will include this experiment to demonstrate the practical value of our probing approach. The current results are preliminary due to time constraints, and a more thorough analysis on more layers/models will be provided in the revised paper.
>
> |     | origin | 1e-6 | 1e-5 | 1e-4 | 1e-3 |
> |-----|-----|-----|-----|-----|-----|
> | TSP100  | 0.57%  | 0.58%  | 0.57%  | 0.57%  | 0.57%  |
> | TSP200 (generalization)   | 0.86%  | 0.88%  | 0.86%  | **0.73%**  | 0.82%  |
> | TSP1000 (generalization)  | 3.17%  | **2.87%**  | 2.93%  | 2.97%  | 3.05%  |
>
> Beyond this, we will enrich our discussions in Section 5 to outline several promising avenues for NCO models to enhance practical implications, including compression, distillation, and pruning of their representation space. Probing NCO layers across CO tasks and training stages may reveal task-specific dimensional requirements, thereby providing a systematic alternative to heuristic selection. We note that while no single study can encompass all possible experiments, our proposed probing pipeline and CS-Probing will serve as powerful tools for future research. We look forward to seeing how it can facilitate the discovery of further actionable insights to advance the field of NCO.
>
> ---
>
> **2. Q1-*While probing helps to build a smaller model with similar performance, same outcome can be achieved by using hyperparameter tuning methods. What other practical implications?***
>
> First, we would like to clarify that regarding your point that "the same outcome can be achieved by using hyperparameter tuning methods," this is somewhat incorrect. We apologize for any misunderstanding. Next, we would like to further explain the practical implications of probing and our proposed CS-Probing method.
>
> I. Why can't hyperparameter tuning methods achieve the outcomes obtained through probing? Specifically, what distinguishes the conclusions drawn from our probing methods from simply reducing the model's dimensionality while maintaining similar performance?
>
> - **Although LEHD relies on only two key dimensions, it cannot simply reduce its embedding dimensions at will.** Although the LEHD model appears to rely primarily on just two dimensions of the final-layer embeddings to make near-equivalent decisions, this does not imply that the entire model operates using only these two dimensions. The earlier layers still require high-dimensional representations to effectively learn and process information.
> - **CS-Probing provides valuable insights that can be used to improve NCO models, which experimental methods like hyperparameter tuning cannot achieve.** In our response above (under "Cons"), we demonstrate how insights obtained through CS-Probing can guide LEHD model improvements in generalization. This represents a concrete application that cannot be achieved through standard experimental practices such as hyperparameter tuning. Methods like hyperparameter tuning can only improve the model heuristically, rather than through the analysis and understanding of the model’s internal mechanisms, which allows for more evidence-based research.
>
> II. Practical implications of probing and our proposed CS-Probing method.
>
> - **Probing enhance transparency and address the key questions raised by the operations research (OR) community.** We uncover what knowledge NCO models capture and how they are leveraged to solve CO problems. These represent foundational questions raised by the OR community concerning the soundness of NCO methods. Our work directly studies them: NCO models capture both low-level features such as Euclidean distance and high-level strategies like avoiding myopic decisions.
> - **In the original manuscript, we have demonstrated how CS-Probing can be used to analyze and understand NCO models, providing actionable insights for improvement.** Through CS-Probing, we find that: (1) distinct inductive biases learned by different NCO models; (2) they encode decision rationales in specific embedding dimensions closely tied to generalization; and (3) in the subspace formed by these key dimensions, model decisions can be interpreted by examining only a few informative dimensions.
>
> III. Conclusion.
>
> To the best of our knowledge, we are the first to introduce probing techniques in the context of NCO. Our proposed probing tasks and the CS-Probing framework not only provide a new method comparable to hyperparameter tuning for improving model performance, but also allow us to verify whether specific knowledge is encoded in NCO representations and to reveal how black-box models make decisions by identifying the most informative embedding dimensions.
>
> This is significant for future research, as it opens up new directions for understanding the inductive biases of NCO models, evaluating the presence and form of knowledge encoded in their embeddings, and guiding the design of more efficient and interpretable representations. As illustrated in the additional experiment presented in our response above (under “Cons”), probing can directly inform actionable strategies. This provides supporting evidence for the practical utility of probing in both interpreting and improving NCO models.
>
> ---
>
> **3. Q2-*For Probing Task 2, how does your study address the issue of non-uniqueness in optimal solutions, especially when drastically different solutions have similar optimization costs?***
>
> The answer to this question can be found in lines 624 to 631 on page 15 of the manuscript. Here, we summarize the relevant text from the manuscript to explain how we address the issue of non-uniqueness in optimal solutions, for your reference:
>
> - We first follow the procedure described in page 614 to 623 of the manuscript to collect positive and negative labels for a given instance. Specifically, we extract an edge from the optimal solution as the positive label (label 1), and the edge connected to the same node with the closest Euclidean distance that is not part of the optimal solution as the negative label (label 0). For example, the positive edge might be between node ***a*** and node ***b***, while the negative edge is between node ***a*** and node ***c***.
> - We then examine the edge labeled as 0. To verify its correctness, we add a constraint to the mathematical model that forces the connection between nodes ***a*** and ***c***. The new optimal solution obtained under this constraint has a higher cost than the original unconstrained solution, indicating that this edge is not part of any optimal solution for this instance.
> - Similarly, for the data labeled as 1, we add a constraint that prevents the connection between nodes ***a*** and ***b***. The resulting solution is also worse than the original, confirming that both labels are valid.
>
> Using the above method, we ensure that the routing instance dataset we collect does not contain instances with multiple optimal solutions, thereby preserving the rigor and validity of the probing dataset.

---

> > ### Comment · Reviewer_MBie · 2025-08-05
> >
> > Thank you for your answer to myself and to other reviewers as well. The new experiments has addressed most of my concerns. I would like to ask you a couple of final questions:
> > a. what was the norm that you have used for the regularisation? Could you provide an exact definition?
> > b. Regarding my hyper-parameter tuning comment. Do I understand it correctly based on your answer to me and the zeroing table provided to another reviewer that by limiting the final layer dimensions to e.g. just two the performance will decrease significantly?

---

> > > ### Author Response · Authors · 2025-08-05
> > >
> > > Thank you for your response and for acknowledging that our rebuttal has addressed most of your concerns. We also sincerely appreciate the time you took to review our discussions with other reviewers, which makes us feel that our work has been worthwhile. Below are our responses to your two questions.
> > >
> > > > a. What was the norm that you have used for the regularisation? Could you provide an exact definition?
> > >
> > > We adopted the standard $L_1$-based sparsity regularization. Specifically, for node $i$ embedding vector $\mathbf{h}_i$ $\in$ $\mathbb{R}^d$, we compute its $L_1$ norm as
> > > $\|\\mathbf{h}\_i\|\_1 = \sum\_{j=1}^{d} |\\mathbf{h}\_{ij}|$
> > > and then take the average over all nodes, where these averaged values are used as the regularization loss, scaled by a coefficient $\lambda$ as we mentioned in the table to control the strength of the regularization. This loss is then directly added to the total loss during training. We will incorporate a detailed description of the regularization approach, along with the additional results that validate the actionable insights derived from probing, into Section 4.4 of the camera-ready version.
> > >
> > > > b. Regarding my hyper-parameter tuning comment. Do I understand it correctly based on your answer to me and the zeroing table provided to another reviewer that by limiting the final layer dimensions to e.g. just two the performance will decrease significantly?
> > >
> > > Your understanding is partially correct, but there seems to be a slight misunderstanding. We would like to take this opportunity to clarify the matter thoroughly. When **LEHD uses only the two key dimensions, 31 and 97, while zeroing out all others**, **the performance does NOT drop "significantly"**. The degradation is minor: the gap from the known optimal solution increases only slightly from 0.58% to 0.65%. These results are shown in Table 3 on page 8 of the manuscript.
> > >
> > > What you observed in the table provided in our response to Reviewer ksMb’s Q3 was the opposite setup: **we zeroed out the two key dimensions (31 and 97) and kept the remaining dimensions**. This **led to a substantial drop in performance**, with the gap exceeding 60%. This means that these two dimensions in the final decision layer are the core contributer to the perofrmance of LEHD and are useful to be levegraed to the generalization while other dimensions are not. This neuron ablation experiment was designed specifically to demonstrate the critical importance of dimensions 31 and 97 in LEHD's embedding.
> > >
> > > **We believe the above discussions demonstrate something that hyperparameter tuning is not able to achieve: identifying key embedding dimensions that are functionally important to the backbone model and linking them to actual decision-relevant knowledge and model behaviors.** For example, as shown in Table 2 of the manuscript, LEHD uses dimensions 31 and 69 to capture Euclidean distances, and dimensions 31 and 97 to extract information related to avoiding myopic decisions. Moreover, we clarify that **hyperparameter tuning remains a valuable technique for finding suitable model architecture to study "what works well", but it is not in conflict with our proposed probing methods which study "why it works".** Instead, they are highly complementary, where probing provides transparent and interpretable insights that can guide more informed hyperparameter adjustments.
> > >
> > >
> > > Lastly, we would like to highlight the research value of our work. Beyond enhancing the interpretability of NCO models, it has the potential to serve as a powerful tool for uncovering actionable insights that can guide and verify future model improvements. For example, in our current work, we show that: how the sparsity insight at the last decision layer of LEHD can be used to improve generalization. We believe a promising future direction is to further use probing to localize more decision-relevant information within embeddings, and then design models specifically tailored to leverage this structure.
> > >
> > > Once again, we sincerely thank you for your time, attention to our work, and the constructive discussions. We will incorporate all our dicussed points into the final version of the paper.

---

### Decision · Program_Chairs · 2025-09-17

**Decision:**

Accept (spotlight)

**Comment:**

Neural combinatorial optimization (NCO) is an emerging and powerful approach for solving classical combinatorial problems. However, while NCO models have shown impressive performance, the internal representations they learn remain largely uninterpretable. This paper addresses this gap by introducing the first method for probing NCO models, revealing several interesting insights into their inner workings.

All the reviewers were enthusiastic about the paper with two of them recommending acceptance and two others recommending weak acceptance. They highlighted several key strengths, including the novel and practical use of probing to understand and improve NCO models, the extensive experimental validation, and the paper's overall clarity and high quality of writing.

Reviewers initially raised a few weaknesses, but the consensus is that the authors have addressed them all effectively in the rebuttal and subsequent discussions. Specifically, the authors clarified the actionable implications by demonstrating that their insights can directly improve model performance, added new experiments on larger-scale problems, and successfully applied their probing method to modern diffusion-based NCO models.

Given that the paper's strengths significantly outweigh any initial concerns, I am pleased to recommend **acceptance**. Furthermore, because this work provides the first look into the internal mechanisms of this important class of models and opens up new avenues for their analysis and improvement, I also recommend that it be highlighted as a **spotlight** presentation.

I strongly encourage the authors to incorporate the new experiments and discussions from the rebuttal into the revised paper.